# Steer Away from Mode Collisions: Improving Composition in Diffusion Models

**Debottam Dutta** *   **Jianchong Chen**   **Rajalaxmi Rajagopalan**
**Yu-Lin Wei**   **Romit Roy Choudhury**
University of Illinois at Urbana-Champaign

## Abstract

We propose to improve multi-concept prompt fidelity in text-to-image diffusion models. We begin with common failure cases—prompts like "a cat and a dog" that sometimes yields images where one concept is missing, faint, or colliding awkwardly with another. We hypothesize that this happens when the diffusion model drifts into mixed modes that over-emphasize a single concept it learned strongly during training. Instead of re-training, we introduce a *corrective sampling strategy* that steers away from regions where the joint prompt behavior overlaps too strongly with any single concept in the prompt. The goal is to steer towards "pure" joint modes where all concepts can coexist with balanced visual presence. We further show that existing multi-concept guidance schemes can operate in unstable weight regimes that amplify imbalance; we characterize favorable regions and adapt sampling to remain within them. Our approach, **CO3** (Concept Contrasting Corrector), is plug-and-play, requires no model tuning, and complements standard classifier-free guidance. Experiments on diverse multi-concept prompts indicate improvements in concept coverage, balance and robustness, with fewer dropped or distorted concepts compared to standard baselines and prior compositional methods. Results suggest that lightweight corrective guidance can substantially mitigate brittle semantic alignment behavior in modern diffusion systems. Code is available at `https://github.com/debottam-dutta7/co3`

## 1 Introduction

Recent diffusion models (Ho et al., 2020; Ramesh et al., 2022a; Rombach et al., 2022) have ushered significant breakthroughs in Text-to-Image (T2I) synthesis, producing high-fidelity images from textual descriptions. However, ensuring the generated images faithfully adhere to the prompt, a challenge known as semantic alignment (Chefer et al., 2023; taihang Hu et al., 2024; Liu et al., 2023), remains a problem. Concretely, for a given prompt $C$, T2I models like StableDiffusion (Rombach et al., 2022) sample from the modes (or high probability regions) of the learned distribution, $p(x \mid C)$. While such models can produce high resolution images in general, every so often, the results are surprisingly misaligned even for very simple prompts containing few concepts, e.g., $C$="a cat and a dog". Diagnosing exactly why this behavior emerges periodically is difficult. It is conceivable that the complex training process in high dimensions, especially in conjunction with text embeddings, creates some problematic modes in $p(x \mid C)$.

We hypothesize that problematic modes in $p(x \mid C)$ arise when they overlap with modes of individual concept distribution $p(x \mid c_i)$. Such an overlap biases the generation toward a single concept $c_i$, reducing the prominence of others. For instance, across images of $c_1 =$ "cat" in the training dataset, a few may have an inconspicuous or partial $c_2 =$ "dog" in the background. This image may still fall under the mode of $p(x \mid C)$. We attribute this to training instabilities and relatively less coverage of multi-concept prompts $C$, which cause the model to assign high probability even to weakly conforming images. Said differently, an image of a big cat and an inconspicuous dog can get assigned high probabilities under $p(x \mid C)$, causing semantic misalignment.

Preventing such problematic modes warrants strict and specialized training paradigms; a difficult task for such large models. However, "curing" them after their occurrence is a more viable approach.

---

* Corresponding author. Email: dd24@illinois.edu

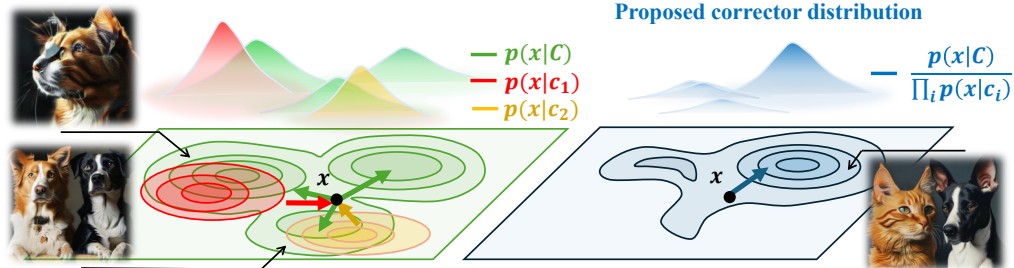

Figure 1: The figure illustrates our hypothesis on mode overlap using a simple 2D toy example. (a) Two modes of the distribution $p(x \mid$ "a cat and a dog"$)$ (in green contour) has significant overlap with the modes of the individual concept distributions $p(x \mid$ "a cat"$)$ (in red contour) and $p_t(x \mid$ "a dog"$)$ (in orange contour). (b) The proposed (unnormalized) corrector distribution $p(x \mid$ "a cat and a dog"$)/(p(x \mid$ "a cat"$)p(x \mid$ "a dog"$))$ suppresses these overlaps, steering the generation away from problematic modes. The arrows indicate the denoising directions.

Assuming our hypothesis is true, we propose a cure for problematic modes. Our intuitive idea is to go away from problematic modes and move towards modes under which none of the individual concepts are strong. To realize this, we propose to design a *corrector* that generates samples from the following distribution:

$$\tilde{p}(x \mid C) \ \propto \ \frac{p(x \mid C)}{\prod_i p(x \mid c_i)}. \tag{1}$$

Figure 1 illustrates the intuition behind our proposal. Our corrector distribution $\tilde{p}(x \mid C)$ assigns low probability to regions where $p(x \mid C)$ overlaps with individual $p(x \mid c_i)$; we deem them as degenerate modes dominated by a single concept. By suppressing these overlaps, the corrector emphasizes *pure* $p(x \mid C)$ modes where all concepts coexist without one overwhelming the others. From a probabilistic perspective, this acts as a corrective factor: while $p(x \mid C)$ may assign high probability to weakly conforming images due to training noise or limited multi-concept data, dividing by the marginals removes this bias and sharpens the distribution toward genuine multi-concept samples. As a result, the modes we target are more semantically aligned and less prone to concept suppression or distortion.

Correction sampling from $\tilde{p}(x \mid C)$ can be achieved by composing scores from constituent component distributions $\nabla_x \log p(x \mid C), \{\nabla_x \log p(x \mid c_i)\}_i$ (Liu et al., 2022). While there are many ways to compose, we analyse and show that composition through weighted sum of Tweedie-means—in the Tweedie denoised space—offers a more general framework that subsumes existing approaches.

In particular, we show that two classes of correction sampling—noise-resampling and latent correction (Bao et al., 2024; Rassin et al., 2023; taihang Hu et al., 2024; Kwon & Ye, 2025)—become special cases of Tweedie-mean composition under different weighting schemes. This allows us to design a hybrid composition framework that serves the purpose of resampling at time $T$, and then toggles to latent correction at later steps. By latent, we mean that the correction steps are accomplished in between each DDIM time step, allowing **CO3** to be a plug-and-play, model-agnostic and gradient-free approach for T2I models. Comparison of **CO3** against SOTA baselines shows stronger semantic alignment to prompts (measured using multiple metrics), giving empirical evidence of our hypothesis.

## 2 BACKGROUND

■ **Conditional generation using Classifier-Free Guidance (CFG).** In diffusion-based Text-to-Image (T2I) generation (Rombach et al., 2022; Saharia et al., 2022a; Ramesh et al., 2022b), given the noisy latent $x_t$ at timestep $t$, a denoised estimate can be derived using Tweedie's formula:

$$\hat{x}_0 = \frac{x_t - \sqrt{1 - \bar{\alpha}_t}\, \epsilon_\theta(x_t, c, t)}{\sqrt{\bar{\alpha}_t}}, \tag{2}$$

where $\epsilon_\theta$ denotes the predicted noise conditioned on the text prompt $c$, and $\bar{\alpha}_t$ is the cumulative product of the noising schedule. This step corresponds to the *denoising* stage, recovering an estimate of the clean signal $x_0$. In the DDIM sampler (Song et al., 2021), under the noise-free condition, the subsequent step deterministically evolves $\hat{x}_0$ to $x_{t-1}$ without introducing additional stochasticity:

$$x_{t-1} = \sqrt{\bar{\alpha}_{t-1}}\, \hat{x}_0 + \sqrt{1 - \bar{\alpha}_{t-1}}\, \epsilon_\theta(x_t, c, t). \tag{3}$$

Here, the same predicted noise $\epsilon_\theta$ is reused, eliminating the renoisification step present in stochastic samplers such as DDPM(Ho et al., 2020). This perspective highlights how the Tweedie decomposition denoising via $\hat{x}_0$, followed by deterministic reconstruction of $x_{t-1}$, is naturally aligned with DDIM sampling.

In practice, most T2I models adopt *classifier-free guidance* (CFG) (Ho & Salimans, 2022), where predictions from both the conditional and unconditional models are combined:

$$\epsilon_t^{\lambda,c} \;=\; \lambda\, \epsilon_\theta(x_t, c, t) \;+\; (1-\lambda)\, \epsilon_\theta(x_t, \varnothing, t) \tag{4}$$

where $\lambda > 1$ controls the guidance strength. $\epsilon_t^{\lambda,c}$ is the composed noise prediction under CFG. Then the denoising and DDIM steps proceed as before, but using $\epsilon_t^{\lambda,c}$ in place of $\epsilon_\theta(x_t, c, t)$.

**Tweedie View with CFG.** Substituting $\epsilon_t^{\lambda,c}$ into the Tweedie denoising and DDIM update yields:

$$\hat{x}_0^{\lambda,c} = \frac{x_t - \sqrt{1 - \bar{\alpha}_t}\, \epsilon_t^{\lambda,c}}{\sqrt{\bar{\alpha}_t}} = \frac{1}{\sqrt{\bar{\alpha}_t}} \hat{x}_{tweedie}[\epsilon_t^{\lambda,c}], \tag{5}$$

$$x_{t-1} = \frac{\sqrt{\bar{\alpha}_{t-1}}}{\sqrt{\bar{\alpha}_t}}\, \hat{x}_{tweedie}[\epsilon_t^{\lambda,c}] + \sqrt{1 - \bar{\alpha}_{t-1}}\, \epsilon_t^{\lambda,c} \tag{6}$$

where $\hat{x}_{tweedie}[\epsilon_t^{\lambda,c}] := x_t - \sqrt{1 - \bar{\alpha}_t}\, \epsilon_t^{\lambda,c}$ is the Tweedie mean from the CFG noise at $t$. Thus, CFG in the Tweedie framework can be interpreted as modifying the denoised estimate $\hat{x}_0$ to $\hat{x}_0^{\lambda,c}$ before the deterministic step to $x_{t-1}$ (Chung et al., 2024; Kwon & Ye, 2023).

■ **Correction-based approaches for conditional generation.** A number of recent works have addressed the challenge of compositional text-to-image generation using correction-based approaches (Chefer et al., 2023; Liu et al., 2023; Bao et al., 2024; Rassin et al., 2023; taihang Hu et al., 2024). During sampling with classifier-free guidance, these methods iteratively correct the latent variable by applying gradient updates of the form

$$x_t^{k+1} = x_t^k - s\, \nabla_{x_t} \mathcal{L}(x_t, c), \quad k = 1, 2, \ldots, M - 1. \tag{7}$$

and then use the final refined latent $x_t^M$ to predict the next DDIM step $x_{t-1}$, using Eqs. 5 and 6. Here $\mathcal{L}(x_t, c)$ is a task-specific loss function that enforces better alignment with the target condition $c$, and $s$ is a step size. This iterative update can be interpreted as correcting the diffusion guidance process using a corrector distribution of the form:

$$\tilde{p}_t(x_t, c) \propto \exp\big(-\mathcal{L}(x_t, c)\big), \tag{8}$$

which refines the generative process at each timestep. In the special case where the step size $s = \sigma_t^2 = 1 - \bar{\alpha}_t$ (the noise scale at timestep $t$), the update rule becomes equivalent to iterative Tweedie-mean correction (see Eq. 5 above), where $x_t^{k+1}$ is reinterpreted as the next estimated Tweedie mean given $x_t^k$ for the distribution $\tilde{p}_t(x_t, c)$.

■ **Composable-diffusion.** Generating samples that satisfy multiple conditions $\{c_1, \ldots, c_K\}$ can be formulated as sampling from the joint distribution

$$\tilde{p}_0(x_0 \mid c_1, \ldots, c_K) \;\propto\; p(x_0) \prod_{k=1}^{K} p_0(c_k \mid x_0). \tag{9}$$

To achieve this, Liu et al. (2022) proposed *composable diffusion*, which directly composes the output scores (predicted noises) from different conditional diffusion models using CFG during sampling.

Specifically, in the text-to-image setting, if the prompt $C$ can be decomposed into constituent concepts $\{c_1, c_2, \ldots, c_K\}$, the outputs of the corresponding diffusion chains can be combined as

$$\tilde{\epsilon}_t^{\lambda,C} \;=\; \epsilon_t^\phi + \lambda_1\big(\epsilon_t^{c_1} - \epsilon_t^\phi\big) + \lambda_2\big(\epsilon_t^{c_2} - \epsilon_t^\phi\big) + \cdots + \lambda_K\big(\epsilon_t^{c_K} - \epsilon_t^\phi\big) \tag{10}$$

where $\epsilon_t^\phi$ denotes the unconditional score, and $\lambda_k$ controls the guidance strength for concept $c_k$. The next sample is then predicted via the Tweedie formulation:

$$x_{t-1} = \frac{\sqrt{\bar{\alpha}_{t-1}}}{\sqrt{\bar{\alpha}_t}} \, \hat{x}_{\text{tweedie}}\left[\tilde{\epsilon}_t^{\lambda,C}\right] \; + \; \sqrt{1 - \bar{\alpha}_{t-1}} \, \tilde{\epsilon}_t^{\lambda,C}. \tag{11}$$

Although this approach is model-agnostic and conceptually simple, its performance is often poor, since the above linear composition of scores is incorrect or doesn't correspond to the score of the diffusion forward distribution $\tilde{p}_t(x_t \mid c_1, \ldots, c_K)$ at any timestep $t > 0$ (Du et al., 2024).

In summary, both the correction-based approach and the composable diffusion idea can be interpreted as different ways of approximating Tweedie-means $\hat{x}_{tweedie}[\epsilon_t^{\lambda,c}]$ at time $t$.

## 3  CO3: CONCEPT CONTRASTING CORRECTOR

We aim to combine the strengths of correction-based approaches and composable diffusion. On one hand, correction-based methods are powerful and explicitly improve compositional alignment, but they are subject to the complex gradient of the base model. On the other hand, composable diffusion is fully model-agnostic, but suffers from poor performance because linear score composition is not consistent with the diffusion forward process Du et al. (2024).

To take advantage of both, assume that prompt $C$ can be decomposed into constituent concepts $\{c_1, c_2, \ldots, c_K\}$. We propose an explicit Corrector distribution, which we call **CO3** (*Concept Contrasting Corrector*), based on our hypothesis on mode overlap discussed in Sec. 1. Specifically, we define the corrector distribution at each timestep $t$ as:

$$\tilde{p}_t(x_t, C) \propto \frac{p_t(x_t \mid C)^{\lambda_0}}{\prod_{k=1}^K p_t(x_t \mid c_k)^{\lambda_k}} \tag{12}$$

where $\lambda_i > 0, i = 0, .., K$ are composition weights. As discussed in Sec. 1, this corrector steers the generation toward regions where the distribution $p_t(x_t \mid C)$ is high, while simultaneously avoiding regions of overlap with the individual concept distributions $\{p_t(x_t \mid c_k)\}_k$. Observe that this encourages the model to generate samples that satisfy all concepts in $C$ without over-emphasizing any single concept. We present the **CO3** corrector pseudo code in Algorithm 1.

To sample from the unnormalized probability distribution in Eq. 12, an well-known approach is to compose the concept distributions $\{p_t(x_t \mid c_k)\}_k$ in the space of score functions (Liu et al., 2022; Du et al., 2024). We break-away from this approach and compose the distribution in the Tweedie mean space; we show how this offers a more general framework for composition. In Tweedie-denoised space we define composition as:

$$\tilde{x}_{tweedie} = w_0 \, \hat{x}_{tweedie}[\epsilon_t^{\lambda,C}] \; + \; w_1 \, \hat{x}_{tweedie}[\epsilon_t^{\lambda,c_1}] \; + \; \ldots \; + \; w_K \, \hat{x}_{tweedie}[\epsilon_t^{\lambda,c_K}] \tag{13}$$

where $w_0 = \lambda_0$ and $w_i = -\lambda_i, i = 1, \ldots, K$ are concept weights. Note: $\hat{x}_{tweedie}[\epsilon_t^{\lambda,c_k}]$ is the Tweedie mean corresponding to the CFG composed noise prediction $\epsilon_t^{\lambda,c_k}$ for concept $c_k$ with guidance weight $\lambda$.

**How should composition weights be chosen?**  We analyze the effect of different weight assignments to Eq. 13, in particular, how the constraint on the concept weights $w_i$ influences both the theoretical interpretation and the empirical behavior of the compositional Tweedie mean.

---

**Algorithm 1** DDIM with **CO3** Corrector

**Require:** $x_T$, number of DDIM steps $T$, timestep threshold $T_c$, prompt $C$, set of concepts $\{c_k\}$
1: **for** $t = T$ down to 1 **do**
2:     **if** $t > T_c$ **then**
3:         $x_t^{(2)} \leftarrow$ **CO3**$(x_t, C, c_1, \ldots, c_k)$
              ▷ Proposed correction
4:         $x_t \leftarrow x_t^{(2)}$
5:     **end if**
6:     $x_t \leftarrow$ DDIM_step$(x_t)$     ▷ Reverse DDIM
7: **end for**

---

A valid compositional Tweedie mean $\tilde{x}_{tweedie}$ in Eq. 13 must be one that can be expressed in the definition given in equation 5 as DDIM goes through a series of Tweedie means for image generation as described in section 2. Proposition 1 shows that this is guaranteed only when the weights satisfy the normalization condition $\sum_i w_i = 1$.

**Proposition 1.** *Let $\hat{x}_{tweedie}[\epsilon_t^{\lambda,c}] := x_t - \sigma_t\,\epsilon_t^{\lambda,c}$ be the tweedie mean from CFG composed noise $\epsilon_t^\lambda = \epsilon_t^\phi + \lambda(\epsilon_t^C - \epsilon_t^\phi)$ for some $\lambda$. Let, $\tilde{\hat{x}}_{tweedie}$ be the composed tweedie-mean defined as $\tilde{\hat{x}}_{tweedie} = \sum_k w_k \hat{x}_{tweedie}[\epsilon_t^{\lambda,c_k}]$. Then,*

- a) ***CO3-corrector:*** *$\tilde{\hat{x}}_{tweedie}$ can be expressed in the form of a tweedie-mean at time $t$, i.e. $\tilde{\hat{x}}_{tweedie} = x_t - \sigma_t\,\tilde{\epsilon}_t^{\tilde{\lambda},C}$ if and only if $\sum_k w_k = 1$. Here $\tilde{\lambda} = \lambda$ and CFG composed noise $\tilde{\epsilon}_t^{\tilde{\lambda},C} = \epsilon_t^\phi + \lambda(\sum_k w_k \epsilon_t^{c_k} - \epsilon_t^\phi)$.*

- b) ***CO3-resampler:*** *$\tilde{\hat{x}}_{tweedie} = -\lambda\,\sigma_t\,\sum_k w_k \epsilon_t^{c_k}$ is weighted noise if and only if $\sum_k w_k = 0$.*

**Remarks:** ❶ See Appendix section A.2 for proof. Proposition 1 a) shows that when $\sum_k w_k = 1$, denoted as ***CO3-corrector***, the composed Tweedie-mean corresponds to the CFG composed noise $\tilde{\epsilon}_t^{\tilde{\lambda},C} = \epsilon_t^\phi + \lambda(\sum_k w_k \epsilon_t^{c_k} - \epsilon_t^\phi)$ with the same guidance scale $\lambda$. This preserves the relative guidance strength between unconditional and conditional scores as proposed by classifier-free guidance (CFG) (see Eq. 4). This is in contrast to composable diffusion methods (Liu et al., 2022; Du et al., 2024) which use arbitrary guidance weights $\{\lambda_i\}$ for composing different concepts (see Eq. 10); hence this does not preserve the CFG form. We believe this explains, at least partly, why composable diffusion often produces out of manifold samples. See Sec. A.3 in Appendix for more discussion.

❷ Proposition 1 b) shows that when the weights sum to zero, denoted as ***CO3-resampler***, composed Tweedie mean doesn't depend explicitly on the current sample $x_t$ and is only a function of the concept noises. Instead of correcting the current sample $x_t$, it replaces $x_t$ with a weighted combination of the concept noises. In other words, when $\sum_k w_k = 0$, the composition is sampling from noise, and only valid at $t = T$ where $x_T \sim \mathcal{N}(0,1)$.

To further characterize the behavior of the composition under the two conditions $\sum_k w_k = 1$ and $\sum_k w_k = 0$, we evaluate the performance of different weighting strategies across timesteps $t$ of the diffusion chain. Figure 2 reveals complementary dynamics between the resampler and the corrector. Specifically, the resampler ($\sum_k w_k = 0$) is effective at high $t$ values near $T$, but its effectiveness diminishes as the denoising process progresses toward lower $t$. Although resampling is not theoretically valid for $t < T$, we observe practical improvements until approximately $t \approx 0.9T$. In contrast, the corrector ($\sum_k w_k = 1$) exhibits gradual improvement with increasing timesteps and saturates around $t \approx 0.75T$. These findings invite a hybrid strategy into **CO3**: apply resampling during early timesteps ($t > T_R$), followed by correction until a threshold ($T_C < t < T_R$), beyond which further gains are marginal.

The pseudocode for the zero-sum-weight resampler (**CO3**-resampler) and the unit-sum-weight (**CO3**-corrector) corrector is provided in Algorithm 2 and Algorithm 3, respectively. Importantly, the same diffusion denoiser model can be reused to compute the individual concept scores under different input conditions $c_k$, thereby *eliminating the need for any costly backward passes to obtain gradients*. Furthermore, the framework operates without reliance on model-specific architectural details, making it *fully model-agnostic* and *gradient-free*.

**Closeness-Aware Concept Weight Modulation:** To make the **CO3** corrector more adaptive, we anchor the weight $w_0$ ($w_0$=1.0 in Algo. 2 and $w_0$ = 2.0 in Algo. 3), and assign weights to each concept based on how *close* the current noise prediction $\epsilon^C$ is, to the noise corresponding to that concept, $\epsilon^{c_k}$. Intuitively, if a sample looks closer to concept $c_k$ than to all others, we want to penalize $c_k$ more strongly (i.e., give it a larger negative weight), while reducing the strength of the other concepts. This encourages the sampler to move away from the nearest mode, preventing collapse toward one dominant concept. Formally, let $d_k = \|\epsilon^C - \epsilon^{c_k}\|$ denote the distance between the current sample and the mode of concept $c_k$. We convert distances $\{d_k\}_{k=1}^K$ into *affinity scores* using an exponential kernel:

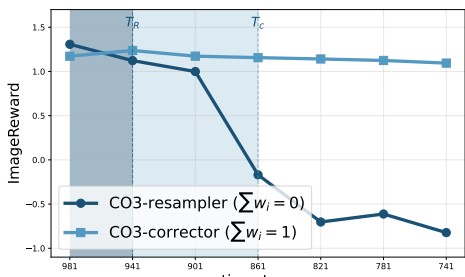

Figure 2: Characterization of Resampler and Corrector steps. Resampling is more powerful at high $t$ while the Corrector improves slowly with more timesteps and saturates.

$$a_k = \exp(-\beta\,d_k), \qquad \beta > 0 \tag{14}$$

**Algorithm 2 CO3-resampler ($\sum w_k = 0$)**

**Require:** $x_t$, denoiser $\psi$, $\bar{\alpha}_t$, $w_{0:K}$, guidance $\lambda$, prompt $C$, concepts $\{c_k\}_k$
1: **for** $p = 1$ to $P$ **do**
2:     $\epsilon^{c_0} = \psi(x_t, C)$, $\epsilon^{c_k} = \psi(x_t, c_k)$, $k = 1, \ldots, K$

3:     $\tilde{\tilde{x}}_{tweedie} = -\lambda\sqrt{1 - \bar{\alpha}_t} \sum_{k=0}^{K} w_k \epsilon^{c_k}$
        ▷ Projection onto noise space

4:     $x_t^{(2)} = \tilde{\tilde{x}}_{tweedie} + \sqrt{1 - \bar{\alpha}_t}\, \epsilon_t^{\phi}$
        ▷ Uncond. manifold correction

5:     $x_t \leftarrow x_t^{(2)}$
6: **end for**
7: **return** $x_t$

**Algorithm 3 CO3-corrector ($\sum w_k = 1$)**

**Require:** $x_t$, denoiser $\psi$, $\bar{\alpha}_t$, $w_{0:K}$, guidance $\lambda$, prompt $C$, concepts $\{c_k\}_k$
1: **for** $p = 1$ to $P$ **do**
2:     $\epsilon^{c_0} = \psi(x_t, C)$, $\epsilon^{c_k} = \psi(x_t, c_k)$, $k = 1, \ldots, K$

3:     $\tilde{\tilde{x}}_{tweedie} = \sum_{k=0}^{K} w_k \hat{x}_{tweedie}[\epsilon_t^{\lambda,c_k}]$
        ▷ Tweedie mean composition

4:     $\tilde{\tilde{x}}_{tweedie} \leftarrow \tilde{\tilde{x}}_{tweedie} * \frac{\|\hat{x}_{tweedie}[\epsilon_t^{\lambda,c_0}]\|_2}{\|\tilde{\tilde{x}}_{tweedie}\|_2}$
        ▷ Length normalization

5:     $x_t^{(2)} = \tilde{\tilde{x}}_{tweedie} + \sqrt{1 - \bar{\alpha}_t}\, \epsilon_t^{\phi}$

6:     $x_t \leftarrow x_t^{(2)}$
7: **end for**
8: **return** $x_t$

so that concepts closer to the current sample (smaller $d_k$) receive higher affinity. These affinity scores are then normalized to define the weights:

$$w_k = -\frac{a_k}{\sum_{j=1}^{K} a_j}, \tag{15}$$

which ensures each weight is negative and the total sum satisfies $\sum_{k=1}^{K} w_k = -1$. As a result, concepts that are closer to the current sample receive stronger negative weights, while farther concepts are down-weighted in proportion to their distance. This weight modulation scheme samples from the whole hyper-plane $\sum_{k=0}^{K} w_k = 1$ (for **CO3**-correction) or $\sum_{k=0}^{K} w_k = 0$ (for **CO3**-resampling) instead of fixed weights for the entire course of generation.

## 4 EXPERIMENTS

### 4.1 EXPERIMENTAL SETUP

**Implementation details:** We implement our method on SDXL (Podell et al., 2023) using a 50-step DDIM sampler with guidance scale $\lambda = 5.0$. **CO3** is applied during the first 20% of denoising steps ($T_c = 0.8T$), with the initial three steps reserved for resampling. For weight modulation, we set $\beta = 0.8$. All experiments are performed on a single NVIDIA A6000 GPU (48GB). Additional implementation details are provided in the Appendix section A.4.

**Evaluation benchmark and metrics:** We first evaluate our method on simpler prompts (two concepts) using the Attend-Excite benchmark from Chefer et al. (2023) which contains categories of prompts: Animal–Animal, Animal–Object, and Object–Object. We then take the best performing baselines from this task and compare them with **CO3** on more complex prompts (three or more concepts) from T2ICompbench Huang et al. (2023) and rare or novel concept prompts from RareBench (Park et al., 2024). Specifically, we consider "Complex" category from T2ICompbench and "Concat", "Relations" and "Complex" from RareBench; the subset of categories which stress multi-concept interactions, attribute entanglement and rare-concept scenarios.

For evaluation, we adopt BLIP-VQA (Huang et al., 2023), ImageReward (Xu et al., 2023) and DINOv2 (Oquab et al., 2023) scores for simpler prompts. While BLIP-VQA and ImageReward (widely used in recent works (taihang Hu et al., 2024; Hu et al., 2024)) measure the faithfulness of generated-images to input-prompts, DINOv2 measures the quality of concepts generated by each model via a classification score. R2F Park et al. (2024) notes that BLIP-VQA can be unreliable in rare-concept scenarios; hence, following their recommendation, we report metrics that more closely reflect human judgement, namely, ImageReward and human evaluation (from 11 participants).

Table 1: Quantitative comparison of different methods on two concept prompts from Chefer et al. (2023). We evaluate the generated images using three metrics: BLIP-VQA, ImageReward and DI-NOv2 scores on categories Animal-Animal(A-A), Animal-Object(A-O) and Object-Object(O-O). The top performing model is highlighted in ▢ and the 2nd best in ▢.

| | Properties | | | BLIP-VQA ↑ | | | ImageReward ↑ | | | DINOv2 ↑ | | |
|---|---|---|---|---|---|---|---|---|---|---|---|---|
| | train-free | grad-free | model-agnostic | A-A | A-O | O-O | A-A | A-O | O-O | A-A | A-O | O-O |
| SD1.5(Rombach et al., 2022) | ✓ | ✓ | - | 0.3239 | 0.5958 | 0.2730 | -0.2733 | 0.4262 | -0.5521 | 0.2163 | 0.2651 | 0.3010 |
| Attend-excite(Chefer et al., 2023) | ✓ | ✗ | ✗ | 0.6980 | 0.7865 | 0.5155 | 0.8244 | 1.2380 | 0.8741 | 0.2245 | 0.2599 | 0.2921 |
| SynGenRassin et al. (2023) | ✓ | ✗ | ✗ | 0.3348 | 0.7689 | 0.6595 | -0.2445 | 0.9838 | 0.7315 | 0.2187 | 0.2606 | 0.2790 |
| Divide & Bind | ✓ | ✗ | ✗ | 0.7201 | 0.8399 | 0.5887 | 0.8499 | 1.2516 | 0.8134 | 0.2029 | 0.2481 | 0.2865 |
| InitNO(Guo et al., 2024) | ✓ | ✗ | ✓ | 0.7264 | 0.7998 | 0.5406 | 1.0082 | 1.3927 | **1.1383** | 0.2252 | 0.2654 | 0.2953 |
| SDXLPodell et al. (2023) | ✓ | ✓ | - | 0.6950 | 0.8654 | 0.4926 | 0.7820 | 1.5574 | 0.6789 | 0.2234 | 0.2618 | 0.2629 |
| Magnet Zhuang et al. (2024) | ✓ | ✗ | ✗ | 0.6782 | 0.8744 | 0.5398 | 0.6450 | 1.5895 | 0.7856 | 0.2122 | 0.2636 | 0.2624 |
| SynGen(SDXL)Rassin et al. (2023) | ✓ | ✗ | ✗ | 0.6816 | 0.8578 | 0.4652 | 0.6998 | 1.5622 | 0.6441 | 0.2221 | 0.2650 | 0.2710 |
| ToMe(taihang Hu et al., 2024) | ✓ | ✗ | ✗ | 0.6257 | 0.8808 | 0.6440 | 0.3895 | 1.5736 | 1.0118 | 0.2226 | 0.2620 | 0.2545 |
| TweediemixKwon & Ye (2025)[1] | ✓ | ✓ | ✓ | 0.7390 | 0.8059 | 0.4683 | 1.0023 | 1.3127 | 0.7959 | 0.2519 | 0.3105 | 0.3237 |
| Comp-diff(Liu et al., 2022) | ✓ | ✓ | ✓ | 0.2846 | 0.5656 | 0.4529 | -1.1399 | -0.2068 | -0.0955 | 0.2497 | 0.2756 | 0.2908 |
| **CO3**(Ours) | ✓ | ✓ | ✓ | 0.7441 | 0.8878 | 0.5146 | 1.2342 | 1.6744 | 1.0158 | 0.2623 | 0.3259 | 0.3283 |

**Comparison methods:** We compare **CO3** against state-of-the-art approaches, including optimization-based correction methods (Attend-Excite (Chefer et al., 2023), SynGen (Rassin et al., 2023), Divide-Bind (Liu et al., 2023), Magnet Zhuang et al. (2024), ToME (taihang Hu et al., 2024)); composable generation methods Comp-Diff (Liu et al., 2022), Tweediemix (Kwon & Ye, 2025); LLM augmented method R2F (Park et al., 2024) and the noise-optimization method InitNO (Guo et al., 2024). For fair comparison against Tweediemix's sampling strategy, we implemented Tweediemix without its LORA-finetuned customization stage. All optimization-based approaches are architecture-dependent and require gradient computations at inference, with InitNO being model-agnostic but still incurring costly backpropagation. In contrast, only Composable Diffusion, Tweediemix and **CO3** are purely sampling-based, model-agnostic, and completely gradient-free.

## 4.2 EXPERIMENTAL RESULTS

Table 2: Quantitative results on multi-concept prompts from T2ICompbench and rare-concepts from RareBench. We report ImageReward (IR) and Human Evaluation (HE) scores.

| | T2ICompbench | | RareBench | | | | | |
|---|---|---|---|---|---|---|---|---|
| | Complex | | Concat | | Relations | | Complex | |
| Method | IR ↑ | HE ↑ | IR ↑ | HE ↑ | IR ↑ | HE ↑ | IR ↑ | HE ↑ |
| SDXL (Podell et al., 2023) | 0.3083 | 0.5983 | 0.1018 | 0.5416 | 0.3607 | 0.5016 | 1.1907 | 0.6283 |
| SynGen(SDXL) (Rassin et al., 2023) | 0.2889 | 0.5133 | 0.0576 | — | 0.3177 | — | 1.1452 | — |
| ToMe (taihang Hu et al., 2024) | 0.2891 | 0.5866 | -0.2041 | 0.4500 | -0.0632 | 0.4983 | 1.0200 | 0.48666 |
| R2F (Park et al., 2024) | 0.3561 | 0.4816 | -0.0162 | 0.4300 | 0.3620 | 0.5200 | 1.1542 | 0.53666 |
| **CO3**(Ours) | 0.4406 | 0.8033 | 0.1610 | 0.6167 | 0.2588 | 0.6200 | 1.2211 | 0.61166 |

**Quantitative Comparison:**

**Simpler prompts (Attend-Excite)**: Table 1 demonstrates the extent to which **CO3**, despite being model-agnostic and optimization-free, competes or outperforms optimization-based and model-agnostic baselines on ImageReward. Gains are substantive in the Animal–Animal and Animal–Object categories. On BLIP-VQA, **CO3** delivers the best performance in these two categories and improves over the base SDXL on the Object category, which contains prompts of the form: "a [color1] [object1] and a [color2] [object2]." Unlike methods such as ToMe taihang Hu et al. (2024), SynGen (Rassin et al., 2023), Attend-Excite (Chefer et al., 2023), and Divide-Bind Liu et al. (2023), which rely on model-specific architectures or explicit attention-based binding losses, **CO3** is architecture-independent, sampling-based, and free of explicit subject–attribute binding. Interestingly, by simply targeting "pure" modes under $p(x \mid C)$, **CO3** discovers layouts that better capture

[1]adapted without the (LORA finetuned) concept-customization stage.

multi-object relations (Animals–Animals, Animal–Objects) and finds high-probability regions that preserve subject–attribute bindings (Objects).

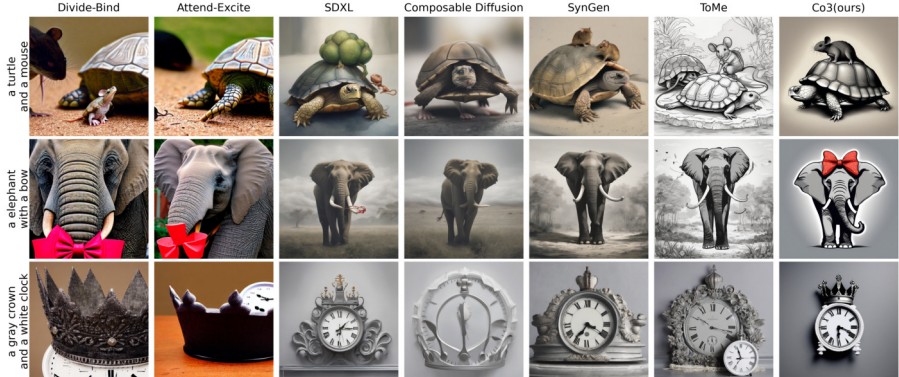

Figure 3: Qualitative comparison of different methods on simpler prompts.

**Multiple and rare-concept prompts (T2ICompbench and RareBench):** Table 2 shows that **CO3** consistently outperforms or matches the best-performing baselines across all categories and metrics. We note that **CO3** is not designed specifically for rare-concept scenarios—it is a general compositional sampling correction that does not rely on rarity-aware priors or LLM-based prompt augmentation. Yet, **CO3** matches or exceeds R2F on RareBench, despite R2F being tailored for rare-concept generation. We believe this indicates that the early-alignment phase of **CO3**, which mitigates concept dominance and encourages sampling from "pure" modes, is naturally beneficial for rare-concept generation as well.

These performance gains empirically validates our hypothesis on mode overlap and also highlights **CO3**'s effectiveness in correcting the compositional limitations of the base model.

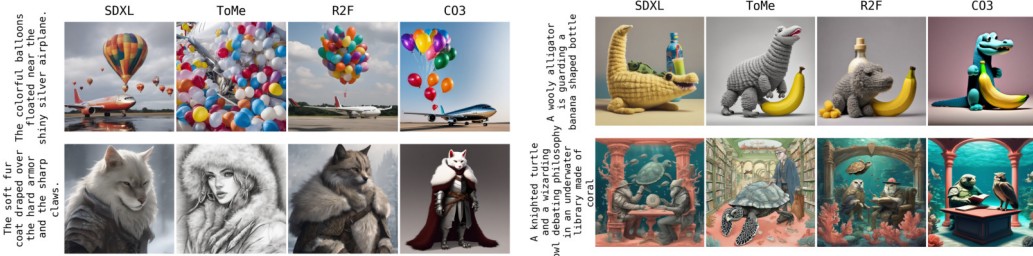

Figure 4: Qualitative comparison of **CO3** with competing methods on complex and rare concept prompts.

Table 3: Ablations study conducted on the **CO3** method. Starting with the base-model SDXL we progressively add different components.

| Method | ImageReward ↑ | | | BLIP-VQA ↑ | | | Avg ↑ |
|---|---|---|---|---|---|---|---|
| | A-A | A-O | O-O | A-A | A-O | O-O | |
| SDXL | 0.7820 | 1.5474 | 0.6789 | 0.6951 | 0.8654 | 0.4926 | 0.8435 |
| + Resampling | 1.0881 | 1.6755 | 0.8263 | 0.7351 | 0.8666 | 0.4528 | 0.9437 |
| + Corrector | 1.0630 | 1.6429 | 0.8949 | 0.7177 | 0.8794 | 0.4796 | 0.9463 |
| + weight-mod. | 1.2342 | 1.6744 | 1.0158 | 0.7441 | 0.8878 | 0.5146 | 1.0118 |
| SDXL+Corrector | 0.9543 | 1.5140 | 0.8894 | 0.7149 | 0.8801 | 0.4989 | 0.9086 |
| + weight-mod. | 1.1732 | 1.5865 | 1.0067 | 0.7211 | 0.8701 | 0.5206 | 0.9797 |

**Qualitative Comparison:** Figure 3 compares **CO3** on Attend-Excite (simpler) prompts with several baselines. We illustrate common issues in compositional alignment—concept missing, attribute mixing, and object binding—across all categories. Divide-Bind (Liu et al., 2023) and Attend-Excite (Chefer et al., 2023) enhance the attention layer of the base model, but often yield cropped or incomplete concepts (rows 1, 3). Comp-Diff (Liu et al., 2022) suffers from missing or merged concepts (rows 2, 3). SynGen (Rassin et al., 2023) and ToMe (taihang Hu et al., 2024) perform well on Object prompts of the form "a [color1] [object1] and a [color2] [object2]," but still exhibit overlapping or mixed concepts (rows 2, 3; columns 5, 6). In contrast, **CO3** achieves stronger object binding in the Animals-Animals and Animals–Objects categories while also improving results on Objects-Objects.

Figure 4 extends qualitative results to complex prompts from the *Complex* category of T2ICompbench and rare-concept prompts from Rarebench Park et al. (2024), comparing **CO3**

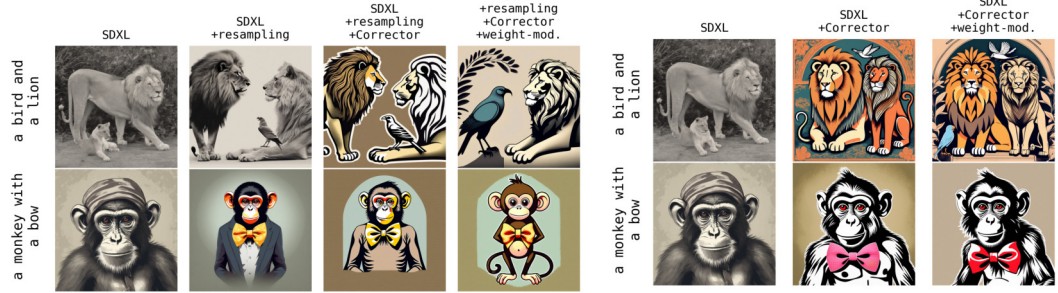

Figure 5: Qualitative results of **CO3** with different configurations on SDXL.

with the top-performing SDXL models (from Table 1). The left half shows prompts from T2ICompbench's Complex category, which involves complex subject-attribute & inter-concept relations. Only **CO3** is able to faithfully generate all concepts and relations, while the other methods often miss concepts or fail to capture relations. The right half shows prompts from RareBench, which involve rare or novel concepts. **CO3** is able to generate extremely rare concepts such as *"banana shaped bottle"* or *"Knighted turtle"*, while the other methods often suffer from training data-bias – generating a regular banana or human knight instead.

**Ablations Studies** are preformed on the key components of **CO3**, starting from the base SDXL model (Table 3 & Figure 5). Adding resampling in the first three DDIM steps substantially improves average performance, particularly for the Animals and Animal–Objects categories. Since image layout is largely determined in the early timesteps, **CO3**-resampler iteratively explores better initial noise configurations that align with the prompt.

Incorporating **CO3**-corrector over the next steps further improves prompts requiring fine-grained details such as color or texture, boosting performance on the Objects category in both ImageReward and BLIP-VQA. Additional gains come from the Closeness-Aware Concept Weight Modulation strategy, which consistently benefits all categories. We hypothesize that as the diffusion trajectory evolves, the latent state often drifts closer to one concept mode $c_i$; the modulation mechanism assigns $c_i$ a more negative weight, repelling the latent and mitigating concept dominance (Tunanyan et al., 2023) commonly observed in stable-diffusion.

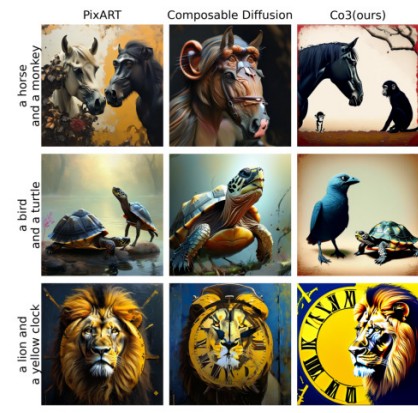

Figure 6: *Model Agnostic behavior*: Qualitative comparison of generation from PixART-$\Sigma$(Chen et al., 2024) base diffusion model, PixART-$\Sigma$ + **CO3**, and PixART-$\Sigma$ + Composable Diffusion (Comp-diff).

Including the Corrector alone already improves performance in categories where attributes must attach to specific objects. This is most clearly visible in Rows 4–5 of the ablation table: prompts of the form "an [attribute1][object1] and [attribute2][object2]" show large gains, because early-timestep dominance (one object absorbing both attributes) is the failure mode that the Corrector resolves. Importantly, the average performance of *SDXL + Corrector + weight-modulation* (last row) is very close to *SDXL + Resampling + Corrector + weight-modulation* (Row 4), which indicates that the Corrector alone is powerful even without Resampling. However, the full improvement is achieved when the Corrector is applied after the Resampling steps.

Table 4: Generalization of CO3 on PixArt and comparison with model-agnostic baselines. cDiff denotes Comp-diff (Liu et al., 2022).

**Comparison with Optimization-Free Approaches:** To further demonstrate **CO3**'s model-agnostic nature and generalization to other base models, we evaluate it on PixArt-$\Sigma$ (Chen et al., 2024) (Table 4). We compare against both the base model and the sampling-based, optimization-free Composable Diffusion (Comp-diff) (Liu et al., 2022). Unlike Comp-diff, which composes diffusion chains within the DDIM prediction step, **CO3** applies composition as a corrector. As shown in Fig. 6, Comp-diff suffers from missing or mixing concepts, while **CO3** consistently preserves all concepts with correct bindings in the output.

| Method | ImageReward ↑ | | | BLIP-VQA ↑ | | |
|---|---|---|---|---|---|---|
| | A-A | A-O | O-O | A-A | A-O | O-O |
| PixArt | 1.066 | 1.697 | 1.023 | 0.691 | 0.861 | 0.559 |
| PixArt + cDiff | 0.917 | 0.047 | 0.339 | 0.208 | 0.591 | 0.472 |
| PixArt + CO3 | 1.380 | 1.770 | 1.081 | 0.746 | 0.864 | 0.581 |

## 5    RELATED WORK

**Optimization-based correction approaches:** *(1) Text-embedding optimization* works like T2I-Zero variants (Tunanyan et al., 2023; Zhuang et al., 2024), and style or prompt inversion (Li et al., 2023a), improve semantic alignment by optimizing or restructuring the text representation during inference. These methods tweak or invert token embeddings so that the denoising trajectory better reflects entity–attribute bindings or multi-concept prompts.

*(2) Attention-map optimization* works (Zhang et al., 2024; Meral et al., 2024; Tumanyan et al., 2023; Hertz et al., 2022) directly manipulate the cross- and self-attention maps during sampling. These methods typically inject constraints or losses that (i) boost token–region correspondence for each entity, (ii) reduce overlap between different concepts, or (iii) preserve early layout information across timesteps. Attend-and-Excite (Chefer et al., 2023) increases the activation of object tokens. Divide-and-Bind (Liu et al., 2023) maximizes total variation to elicit distinct excitations for multiple objects and aligns attribute–entity attention maps, while A-STAR (Agarwal et al., 2023) further reduces cross-token overlap and preserves early attention signals. SynGen (Rassin et al., 2023) leverages syntactic parses to penalize mismatched attention overlaps, ensuring linguistic binding between entities and modifiers. InitNO (Guo et al., 2024) optimizes the initial noise so that sampling begins in more favorable regions that yield stronger, less-conflicted attention.

**Composable generation works:** These works view conditional diffusion models as energy/score functions, enabling algebraic composition. Composed-Diffusion (Liu et al., 2022) frames score-composition under CFG and demonstrates test-time generalization, but suffers from concept mixing and missing. Follow-ups propose training-free and model-agnostic methods like energy-parameterized diffusion and Metropolis/MCMC-corrected samplers that markedly improve multi-condition generation (Du et al., 2024), but their performance is often poor (Chefer et al., 2023; Feng et al., 2022). On a different line, while Skreta et al. (2025) proposes a density estimation approach for composing diffusion chains for concept interpolation, Park et al. (2024) interpolates distributions of frequent and rare-concepts for improved compositional generation. Kwon & Ye (2025) uses a composition similar to our resampler but not in the context of a corrector. Instead they use multiple DDIM forward-backward steps to sample initial noise.

**Layout augmentation image generation:** *(1) Layout-to-image methods:* (Xie et al., 2023; Phung et al., 2024; Kim et al., 2023; Zhao et al., 2023) use a strategy with explicit spatial priors—-boxes, masks, or region texts to bridge text and image: training-free controllers steer attention so objects materialize in designated regions. Other works add instance-level handles, for fine-grained placement and attributes across multiple entities (Wang et al., 2024). In parallel, fine-tuning approaches inject layout channels into the backbone (Li et al., 2023b; Mou et al., 2023; Zhang et al., 2023).

*(2) LLM-augmented methods:* Collectively, these works leverage LLM reasoning/representations to better bridge linguistic structure and the denoising trajectory by (i) decomposing complex prompts into regional sub-tasks (Yang et al., 2024; Hu et al., 2024) (ii) infer layouts from text (Qu et al., 2023), and (iii) act as stronger text encoders or timestep-aware adapters (Saharia et al., 2022b).

## 6    CONCLUSION & FUTURE WORK

We propose a gradient-free, model-agnostic image composition method that switches between noise-resampling and correction. Our *Concept Contrasting Corrector* is based on the hypothesis that composition is degraded by "problematic" modes—those that overlap with individual concept modes, resulting in strong alignment with particular concepts while suppressing others. Our corrector employs a Closeness-Aware Weight Modulation scheme that emphasizes "pure" modes where all concepts coexist without any dominating ones. Crucially, we attribute shortcomings of past composition methods to the choice of composition weights and show that only when the weights sum to 1, the interpretation of Tweedie-mean correction and the CFG guidance strength are preserved. Results demonstrate that **CO3** outperforms baseline approaches for several diffusion model choices and serves as a *plug-and-play* module offering improved compositional generation performance.

While our corrector suppresses concept overlap, the model's predicted score landscape remains biased by training data quality, occasionally leading to poor alignment in unrealistic prompts (Fig. 8). While our work offers a significant boost to multi-concept compositional generation, failure cases still persist and we believe advanced energy-based compositional samplers that explicitly model landscape probabilities can further mitigate these issues. We intend to investigate these edge cases and explore applications beyond image compositional generation in future work.

ACKNOWLEDGMENTS

Acknowledgments: We thank the anonymous reviewers for their valuable feedback. This work was partially supported by NSF #2008338, #1909568, #2148583, and #MRI-2018966. This work used DELTA at NCSA through allocation CIS230230 from the Advanced Cyberinfrastructure Coordination Ecosystem: Services & Support (ACCESS) program, which is supported by U.S. National Science Foundation grants #2138259, #2138286, #2138307, #2137603, and #2138296

REPRODUCIBILITY STATEMENT

All method implementations, inference pipelines are included in the supplementary material. This includes instructions for environment setup, dependencies, and reproducible random seeds. The datasets used in our experiments are publicly available. We provide detailed descriptions of the dataset in Sec. 4. Hyperparameters, inference schedules, and evaluation pipelines are described in Sec. 4, with further details in the Appendix A.4.

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

# A APPENDIX

## A.1 BROADER IMPACTS

**CO3** improves compositional text-to-image generation by adjusting the sampling distribution of diffusion models. At the same time, it introduces certain risks. The method could be misused to create deceptive or misleading visuals, contributing to the spread of misinformation. When applied to depictions of public figures, it may compromise personal privacy. Moreover, the automatically produced content can raise concerns around copyright and intellectual property.

## A.2 COMPOSITION IN TWEEDIE-DENOISED SPACE FOR A VALID CFG

**Classifier free guidance:** In CFG, to sample from $p(x|c)$, we compose the conditonal and unconditonal predicted noise at each time step $t$ as:

$$\epsilon_t^{\lambda,c} = \epsilon_t^\phi + \lambda(\epsilon_t^c - \epsilon_t^\phi) \tag{16}$$

where $\epsilon_t^c$ is the noise predicted for the distribution $p_t(x_t|c)$ at time $t$, i.e., $\nabla_{x_t} \log p_t(x_t|c) = -\frac{\epsilon_t^c}{\sqrt{1-\bar{\alpha}_t}}$. Then we use this composed noise to sample the next step as,

$$\hat{x}_{t,0}^{\lambda,c} = \frac{x_t - \sqrt{1-\bar{\alpha}_t}\epsilon_t^{\lambda,c}}{\bar{\alpha}_t}, \quad x_{t-1} = \sqrt{\bar{\alpha}_{t-1}}\hat{x}_{t,0}^{\lambda,c} + \sqrt{1-\bar{\alpha}_{t-1}}\epsilon_t^{\lambda,c} \tag{17}$$

Let, $\hat{x}_{tweedie}^{\lambda,c} := x_t - \sqrt{1-\bar{\alpha}_t}\epsilon_t^{\lambda,c}$ be the tweedie mean from CFG composed noise at time $t$. Then we can rewrite the above equation as,

$$x_{t-1} = \frac{\sqrt{\bar{\alpha}_{t-1}}}{\sqrt{\bar{\alpha}_t}}\hat{x}_{tweedie}^{\lambda,c} + \sqrt{1-\bar{\alpha}_{t-1}}\epsilon_t^{\lambda,c} \tag{18}$$

**Composition of CFG scores:** (Liu et al., 2022) introduced the idea composable-diffusion where they compose scores from different diffusion models or conditional distributions to generate images from the composed distribution. They assume that, $p(x_0|C) = \prod p(x_0|c_i)$ and proposed the modified CFG composition for each $t$ as,

$$\tilde{\epsilon}_t^{\lambda,C} = \epsilon_t^\phi + \lambda_i(\sum_i \epsilon_t^{c_i} - \epsilon_t^\phi) \tag{19}$$

The issue with above composition is that for arbitrary $\lambda_i$, the composed CFG noise doesn't satisfy the CFG equation in equation 16, i.e., $\tilde{\epsilon}_t^{\lambda,C} \neq \epsilon_t^{\lambda,C}$ for any $\lambda$ in equation equation 16 where $\epsilon_t^{\lambda,C}$ is the noise predicted for the distribution $p_t(x_t|C)$ at time $t$.

To conretize this, we state the following simple result.

**Lemma 1.** *Let's define CFG as a function at time $t$ as, $f_{CFG}(\epsilon_t; \lambda) = \epsilon_t^\phi + \lambda(\epsilon_t - \epsilon_t^\phi)$. Then for any $\lambda$ and $K > 1$, $\sum_k w_k f_{CFG}(\epsilon_t^k; \lambda) = f_{CFG}(\sum_k w_k \epsilon_t^k; \lambda)$ only if $\sum_k w_k = 1$.*

*Proof.* Let, $\tilde{\epsilon}_t^\lambda = \sum_k w_k f_{CFG}(\epsilon_t^k; \lambda)$. Then,

$$\tilde{\epsilon}_t^\lambda = \sum_k w_k(\epsilon_t^\phi + \lambda(\epsilon_t^k - \epsilon_t^\phi))$$

$$= \sum_k w_k \epsilon_t^\phi + \lambda(\sum_k w_k \epsilon_t^k - \sum_k w_k \epsilon_t^\phi)$$

$$= \epsilon_t^\phi \sum_k w_k + \lambda(\sum_k w_k \epsilon_t^k - \epsilon_t^\phi \sum_k w_k)$$

$$= \epsilon_t^\phi + \lambda(\sum_k w_k \epsilon_t^k - \epsilon_t^\phi) \quad \text{if } \sum_k w_k = 1$$

$$= f_{CFG}(\sum_k w_k \epsilon_t^k; \lambda) \quad \text{if } \sum_k w_k = 1$$

If $\sum_k w_k \neq 1$, then $\tilde{\epsilon}_t^\lambda \neq f_{CFG}(\sum_k w_k \epsilon_t^k; \lambda)$. $\square$

**Remark:** One immediate implication of this lemma is that, whenever we sample from a composed distribution with the CFG composed score of the form $\tilde{\epsilon}^\lambda = w_1 \epsilon^{\lambda,1} + w_2 \epsilon^{\lambda,2} + \cdots + w_K \epsilon^{\lambda,K}$, then $\tilde{\epsilon}^\lambda$ corresponds to the CFG noise for the distribution $p(x|C) = \prod_i p(x \mid c_i)^{w_i}$ only when $\sum_i w_i = 1$.

**Proposition 2.** *(Restated) Let $\hat{x}_{tweedie}[\epsilon_t^{\lambda,c}] := x_t - \sigma_t \epsilon_t^{\lambda,c}$ be the tweedie mean from CFG composed noise $\tilde{\epsilon}_t^\lambda = \epsilon_t^\phi + \lambda(\epsilon_t^C - \epsilon_t^\phi)$ for some $\lambda$. Let, $\tilde{\hat{x}}_{tweedie}$ be the composed tweedie-mean defined as $\tilde{\hat{x}}_{tweedie} = \sum_k w_k \hat{x}_{tweedie}[\epsilon_t^{\lambda,c_k}]$. Then,*

*   *a)* **CO3-corrector:** *$\tilde{\hat{x}}_{tweedie}$ can be expressed in the form of a tweedie-mean at time $t$, i.e. $\tilde{\hat{x}}_{tweedie} = x - \sigma_t \tilde{\epsilon}_t^{\tilde{\lambda},C}$ if and only if $\sum_k w_k = 1$. Here $\tilde{\lambda} = \lambda$ and CFG composed noise $\tilde{\epsilon}_t^{\tilde{\lambda},C} = \epsilon_t^\phi + \lambda(\sum_k w_k \epsilon_t^{c_k} - \epsilon_t^\phi)$.*

*   *b)* **CO3-resampler:** *$\tilde{\hat{x}}_{tweedie} = -\lambda \sigma_t \sum_k w_k \epsilon_t^{c_k}$ is weighted noise if and only if $\sum_k w_k = 0$.*

*Proof.* a) The proof can be obtained by simplifying the $\hat{x}_{tweedie}^{comp}$ expression and using direct application of Lemma 1. We have,

$$\tilde{\hat{x}}_{tweedie} = \sum_k w_k \hat{x}_{tweedie}^{\lambda,c_k}$$

$$= \sum_k w_k(x_t - \sigma_t \epsilon_t^{\lambda,c_k})$$

$$= x_t \sum_k w_k - \sigma_t \sum_k w_k \epsilon_t^{\lambda,c_k}$$

$$= x_t \sum_k w_k - \sigma_t \sum_k f_{CFG}(\epsilon_t^{c_k}) \quad (\text{ from Lemma 1})$$

$$= x_t - \sigma_t f_{CFG}(\sum_k w_k \epsilon_t^{c_k}) \quad \text{if} \sum_k w_k = 1$$

So, we have $\tilde{\hat{x}}_{tweedie} = x_t - \sigma_t \tilde{\epsilon}_t^{\lambda,C}$ where $\tilde{\epsilon}_t^{\lambda,C} = f_{CFG}(\sum_k w_k \epsilon_t^{c_k}) = \epsilon_t^\phi + \lambda(\sum_k w_k \epsilon_t^{c_k} - \epsilon_t^\phi)$. If $\sum_k w_k \neq 1$, then $\tilde{\hat{x}}_{tweedie}$ cannot be expressed in the form $x_t - \sigma_t \tilde{\epsilon}_t^{\lambda,C}$ for any $\lambda$ and $\tilde{\epsilon}_t^{\lambda,C}$.

b) From the 4-th equality of part a)

$$\tilde{\hat{x}}_{tweedie} = x_t \sum_k w_k - \sigma_t \sum_k w_k \epsilon_t^{\lambda,c_k}$$

$$= -\sigma_t \lambda \sum_{k=1}^K w_k \epsilon_t^k \text{ (from Lemma 1)}$$

$\square$

### A.3 COMPARISON OF COMPOSITIONS IN SCORE SPACE

In this section we compare our composition framework with Composable-Diffusion. As already described in section 3, we propose composition in Tweedie-denoised space as

$$\tilde{x}_{tweedie} = w_0 \hat{x}_{tweedie}[\epsilon_t^{\lambda,C}] + w_1 \hat{x}_{tweedie}[\epsilon_t^{\lambda,c_1}] + \ldots + w_K \hat{x}_{tweedie}[\epsilon_t^{\lambda,c_K}] \qquad (20)$$

$$(21)$$

which leads to

$$\tilde{x}_{tweedie} = x - \sigma_t \tilde{\epsilon}_t^{\tilde{\lambda},C} \text{ iff } \sum w_{k=0}^K = 1 \qquad (22)$$

$$\text{where } \tilde{\epsilon}_t^{\tilde{\lambda},C} = \epsilon_t^\phi + \lambda(\sum_k w_k \epsilon_t^{c_k} - \epsilon_t^\phi) \qquad (23)$$

Thus, when mixing scores from different conditional distributions, the above condition need to be satisfied for $\tilde{x}_{tweedie}$ to correspond to any coherent joint conditional model. Contrast this with the Composable-Diffusion's (Liu et al. (2022)) noise/score composition:

$$\tilde{\epsilon}_{t,compdiff}^{\lambda,C} = \epsilon_t^{\phi} + \lambda_1\left(\epsilon_t^{c_1} - \epsilon_t^{\phi}\right) + \lambda_2\left(\epsilon_t^{c_2} - \epsilon_t^{\phi}\right) + \cdots + \lambda_K\left(\epsilon_t^{c_K} - \epsilon_t^{\phi}\right) \tag{24}$$

The above composition allows arbitrary weights, which in general violate the unit-sum constraint. In such cases, the result is simply a heuristic linear combination of scores, and the resultant *Tweedie mean isn't valid* and doesn't correspond to any well-defined distribution under the given forward process.

By contrast, CO3 composes Tweedie-means from different conditional distributions. Lemma 1 shows that these weights have to satisfy the same unit-sum condition as above for the resultant composed Tweedie-mean to be a valid one. CO3 maintains this condition and this preserves the update on the Tweedie manifold of the underlying sampler.

## A.4 More Implementation Details

### A.4.1 **CO3** Implementation Details

For the results in Table 1 we use $T_c = 10$(number of time-steps to correct), $T_r = 3$(number of resampling steps) and $P = 5$(number of corrector iteration). For concept aware weight modulation, we use exponential kernel with $\beta = 0.8$ while anchoring $w_0$ at 1.0 for **CO3**-resampler and 2.0 for **CO3**-corrector.

We use Stanza (Qi et al., 2020) to parse the prompts. We parse the prompts to extract different noun chunks and filter each of them to remove articles and adjectives. The remaining proper noun is used as concept in **CO3**. For example, if $C$ is "a black cat and a brown dog", we consider $c_1 =$"cat" and $c_2 =$"dog".

### A.4.2 Baseline Methods

Attend-Excite (Chefer et al., 2023), Divide-Bind (Liu et al., 2023), InitNo (Guo et al., 2024), SynGen (Rassin et al., 2023) and Composable Diffusion (Liu et al., 2022) are methods based on SD1.5. We run experiments on these models using their publicly available code. Tweediemix (Kwon & Ye, 2025), Magnet Zhuang et al. (2024), R2F (Park et al., 2024) and ToMe (taihang Hu et al., 2024) are SDXL based we use their publicly available code-base implementation. We also adapted Composable Diffusion and SynGen to SDXL.

### A.4.3 Additional Human Evaluation

We conducted new human-evaluation studies (with 11 participants) across all benchmarks considered in this paper. For evaluation, we follow the protocol from R2F Park et al. (2024). For each prompt, images from **CO3** and baseline methods are displayed side by side to enable direct comparison. Method names are anonymized and their order is randomly shuffled to reduce bias. Participants provide ratings in $\{0, 1, 2, 3, 4, 5\}$, which are then normalized to $[0, 1]$. Results are shown below (**higher is better**). Results on T2ICompbench and RareBench are included in Table 2 of the main text. Table 5 shows results on Attend-Excite prompts.

Table 5: Human Evaluation results on Attend-Excite prompts. Higher the score, the better.

| Method | Animals | Animals-Objects | Objects |
|---|---|---|---|
| SDXL | 0.5266 | 0.6050 | 0.7083 |
| SynGen | 0.3667 | 0.4366 | 0.7183 |
| Magnet | 0.3700 | 0.5683 | 0.6849 |
| ToMe | 0.4383 | 0.6083 | 0.6166 |
| CO3(ours) | 0.8817 | 0.7466 | 0.8250 |

## A.5 Generalization to higher-order solvers and low inference steps regimes

In addition to the main 50-step SDXL with DDIM experiments, we include results using both (i) low-step DDIM and (ii) low-step high-order DPM++ solvers (10–20 steps).

Please note that, since DDIM sampling has direct connections with denoising and generation via the Tweedie mean (Section 2 in the main paper), **CO3** applies naturally to DDIM. However, our experiments below show that **CO3** continues to provide consistent improvements under higher-order solvers and aggressive sampling budgets as shown in Table 6 & Table 7.

Table 6: Performance of **CO3** with DPM++ solver and in the low inference steps regime.

| Method | Steps | ImageReward ↑ | | | BLIP-VQA ↑ | | | Avg ↑ |
| | | Animals | Animals Objects | Objects | Animals | Animals Objects | Objects | |
|---|---|---|---|---|---|---|---|---|
| SD-DDIM | 50 | 0.7820 | 1.5574 | 0.6789 | 0.6950 | 0.8658 | 0.4925 | 0.8452 |
| CO3-DDIM | 50 | 1.2341 | 1.6743 | 1.0158 | 0.7441 | 0.8878 | 0.5146 | 1.0112 |
| SD-DPM++ | 20 | 0.7089 | 1.5652 | 0.6757 | 0.6974 | 0.8537 | 0.4984 | 0.8332 |
| CO3-DPM++ | 20 | 1.1424 | 1.5249 | 0.9537 | 0.7382 | 0.8842 | 0.4944 | 0.9563 |
| SD-DPM++ | 10 | 0.6834 | 1.5074 | 0.6681 | 0.6768 | 0.8537 | 0.4974 | 0.8145 |
| CO3-DPM++ | 10 | 1.0371 | 1.4520 | 0.9206 | 0.7331 | 0.8765 | 0.4443 | 0.9106 |

Table 7: Performance of **CO3** with DDIM solver in the low inference steps regime.

| Method | Steps | ImageReward ↑ | | | BLIP-VQA ↑ | | | Avg ↑ |
| | | Animals | Animals Objects | Objects | Animals | Animals Objects | Objects | |
|---|---|---|---|---|---|---|---|---|
| SD w/ DDIM | 50 | 0.7820 | 1.5574 | 0.6789 | 0.6950 | 0.8658 | 0.4926 | 0.8452 |
| CO3 w/ DDIM | 50 | 1.2341 | 1.6743 | 1.0158 | 0.7441 | 0.8878 | 0.5146 | 1.0112 |
| SD w/ DDIM | 20 | 0.6965 | 1.4567 | 0.5516 | 0.6832 | 0.8620 | 0.4947 | 0.7907 |
| CO3 w/ DDIM | 20 | 0.8532 | 1.6440 | 0.7481 | 0.6698 | 0.8611 | 0.4315 | 0.8671 |
| SD w/ DDIM | 10 | 0.4906 | 1.1503 | 0.1131 | 0.6428 | 0.8378 | 0.4752 | 0.6183 |
| CO3 w/ DDIM | 10 | 0.5872 | 1.5900 | 0.7117 | 0.6294 | 0.8589 | 0.4453 | 0.8037 |

## A.6 PROMPTS WITH AND WITHOUT BACKGROUNDS

CO3 is strongly concept obedient. Many CO3 generated images from Attend-Excite prompts seem to lack background details. To investigate this behavior, we added background concepts to the same prompts shown in Figure 6 of the main text. Figure 7 shows the resulated images from this experiment. We observe that, CO3 tends to strongly obey the concepts specified in the prompt; when the background is not specified it is conservative with hallucinating the background. However, if the background is explicitly specified, CO3 brings the same degree of richness and quality to both the foreground and the background.

## A.7 TIME AND MEMORY COMPLEXITY

Table 8: Runtime and memory comparison between SDXL, ToMe, and CO3 at different sampling budgets.

| Method | ImageReward ↑ | | | BLIP-VQA ↑ | | | Avg ↑ | Time(s) ↓ | Memory ↓ |
| | Animals | Animals Objects | Objects | Animals | Animals Objects | Objects | | | |
|---|---|---|---|---|---|---|---|---|---|
| SDXL (50 steps) | 0.4906 | 1.1503 | 0.1131 | 0.6428 | 0.8378 | 0.4752 | 0.6183 | 7.20 | 8.5GB |
| ToMe (50 steps) | 0.3894 | 1.5736 | 1.0117 | 0.6257 | 0.8807 | 0.6440 | 0.8542 | 16.58 | 18GB |
| CO3 (50 steps) | 1.2341 | 1.6743 | 1.0158 | 0.7440 | 0.8877 | 0.5145 | 1.0120 | 19.90 | 9.5GB |
| CO3 (20 steps) | 0.8532 | 1.6440 | 0.7481 | 0.6698 | 0.8611 | 0.4315 | 0.8679 | 10.65 | 9.5GB |

We compare **CO3** with its most competitive baseline ToMe taihang Hu et al. (2024) in terms of runtime peak memory and time requirement to generate one sample in Table 8. The experiments are conducted on a NVIDIA A100 gpu.

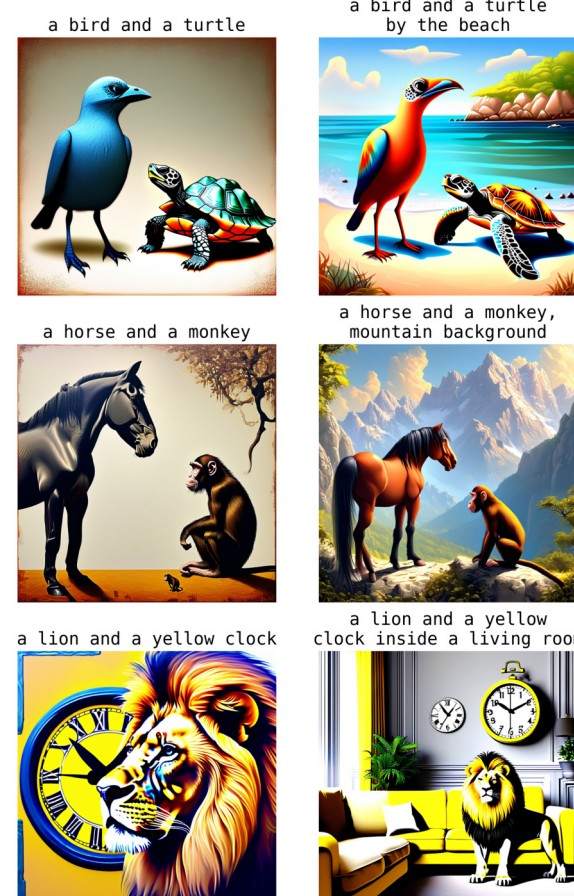

Figure 7: Concept obedient behavior of **CO3**: CO3 tends to strongly obey the concepts specified in the prompt; when the background is not specified it is conservative with hallucinating the background.

Although **CO3** takes more time compared to ToMe with 50 time-steps, it requires significantly less memory. **CO3** is lightweight, and its peak memory usage is only slightly above that of the base SDXL model. ToMe, being a gradient-based method, has substantial memory requirements.

We also emphasize that **CO3** is model-agnostic and can be applied to *any* diffusion-based model. In contrast, ToMe is model-architecture dependent, as its attribute-binding mechanism operates by manipulating internal attention activations of the base model.

For a fair comparison, we additionally tested **CO3** with 20 inference steps. As shown in the table below, the average performance of CO3 with 20 steps remains comparable to, and in many cases better than, ToMe with 50 steps. **CO3** with 20 steps takes 10.65 seconds, which is faster than ToMe.

## A.8    ABLATIONS ON HYPERPARAMETERS

In this section, we analyze the contribution of the following five factors to the performance of our **CO3** corrector.

Notation recap: $\beta$ is the exponential decay factor in the *affinity_scores*; $\lambda$ scales the *composed_score* for CFG; num_resampling is the number of resampling steps at the start of diffusion; num_ts is the number of early timesteps where the corrector is used; num_steps is the number of iterations per corrector application.

### A.8.1 NUMBER OF RESAMPLING STEPS

Table 9 shows results for different resampling steps. We observe that using more resampling steps can blur concept separation early in the diffusion process rather than helping to disentangle concepts.

Table 9: **num_resampling** sweep. Within each group, only num_resampling changes; all other settings are identical.

| Frozen settings | num_resampling | ImageReward (↑) | | | BLIP-VQA (↑) | | |
|---|---|---|---|---|---|---|---|
| | | Animals | Animals_Objects | Objects | Animals | Animals_Objects | Objects |
| $\beta = 0.8$, *num_ts=7*, $\lambda = 0.8$, *num_steps=5* | 3 | 1.2218 | 1.7020 | 0.9405 | 0.7375 | 0.8811 | 0.4782 |
| | 4 | 1.1705 | 1.7031 | 0.9512 | 0.7259 | 0.8828 | 0.4696 |
| $\beta = 0.3$, *num_ts=7*, $\lambda = 0.8$, *num_steps=5* | 3 | 1.1484 | 1.6915 | 0.9005 | 0.7240 | 0.8810 | 0.4691 |
| | 4 | 1.1297 | 1.6948 | 0.8996 | 0.7145 | 0.8774 | 0.4697 |
| $\beta = 1.1$, *num_ts=7*, $\lambda = 0.8$, *num_steps=5* | 3 | 1.2566 | 1.7062 | 0.9916 | 0.7422 | 0.8813 | 0.4834 |
| | 4 | 1.1927 | 1.7164 | 0.9934 | 0.7292 | 0.8835 | 0.4811 |

### A.8.2 CORRECTOR APPLICATION TIMESTEPS

In **CO3**, the corrector is applied during the first num_ts diffusion steps. Table 10 shows that larger num_ts generally yields better results, especially for *animals* and *objects*. This indicates the corrector remains beneficial beyond only the earliest timesteps.

Table 10: **num_ts** sweep with all other settings fixed: $\beta = 1.1$, num_resampling=3, num_steps=5, $\lambda = 0.8$. (Best values are bold)

| num_ts | ImageReward (↑) | | | BLIP-VQA (↑) | | |
|---|---|---|---|---|---|---|
| | Animals | Animals_Objects | Objects | Animals | Animals_Objects | Objects |
| 4 | 1.1862 | 1.7136 | 0.8885 | 0.7430 | 0.8783 | 0.4630 |
| 6 | 1.2510 | **1.7117** | 0.9750 | 0.7432 | 0.8801 | 0.4812 |
| 7 | 1.2566 | 1.7062 | 0.9916 | 0.7422 | 0.8813 | 0.4834 |
| 10 | **1.3149** | 1.6816 | **1.0095** | **0.7576** | **0.8832** | **0.5074** |

### A.8.3 CORRECTOR ITERATIONS

Table 11 reports the effect of increasing the number of corrector iterations. Despite setting num_ts=4, raising num_steps reduces performance substantially, suggesting that applying the corrector only at the very beginning is not sufficient; employing it until much later in the diffusion trajectory is more effective.

Table 11: **num_steps**. Within each group, only num_steps changes; all other settings are identical.

| Fixed settings | num_steps | ImageReward (↑) | | | BLIP-VQA (↑) | | |
|---|---|---|---|---|---|---|---|
| | | Animals | Animals_Objects | Objects | Animals | Animals_Objects | Objects |
| *(a)* $\beta = 1.1$, *num_ts=4*, | 7 | **1.2622** | **1.7129** | **0.9959** | **0.7542** | **0.8843** | **0.4707** |
| *num_resampling=2*, | 10 | 1.2452 | 1.6994 | 0.9753 | 0.7448 | 0.8809 | 0.4706 |
| $\lambda = 0.9$ | 15 | 1.2151 | 1.6701 | 0.9437 | 0.7318 | 0.8781 | 0.4556 |
| *(b)* $\beta = 1.1$, *num_ts=4*, | 7 | **1.2755** | **1.7049** | **0.9424** | **0.7538** | 0.8762 | 0.4611 |
| *num_resampling=2*, | 10 | 1.2503 | 1.6969 | 0.9191 | 0.7446 | **0.8798** | **0.4617** |
| $\lambda = 0.8$ | 15 | 1.2480 | 1.6658 | 0.9063 | 0.7356 | 0.8750 | 0.4577 |

### A.8.4 EXPONENTIAL DECAY FACTOR $\beta$ OF *affinity scores*

We vary $\beta$ in equation 14 from 0.3 to 1.1. Tables 12 and 13 demonstrate that the performance generally increases with larger $\beta$, with the strongest gains in the *animals* and *objects* categories, while a few settings exhibit minor regressions.

Table 12: **Exponential decay factor** $\beta$ with all other settings fixed: `num_resampling=3`, `num_steps=5`, $\lambda = 0.8$. We report two blocks: `num_ts=6` and `num_ts=7`. (Best values for each block are bold)

| num_ts | $\beta$ | ImageReward (↑) | | | BLIP-VQA (↑) | | |
|---|---|---|---|---|---|---|---|
| | | Animals | Animals_Objects | Objects | Animals | Animals_Objects | Objects |
| | 0.3 | 1.1458 | 1.7016 | 0.8801 | 0.7262 | 0.8793 | 0.4613 |
| | 0.6 | 1.1945 | 1.7069 | 0.9010 | 0.7317 | 0.8786 | 0.4645 |
| | 0.7 | 1.2043 | 1.7109 | 0.9152 | 0.7310 | 0.8800 | 0.4706 |
| 6 | 0.8 | 1.2157 | 1.7024 | 0.9156 | 0.7398 | 0.8776 | 0.4751 |
| | 0.9 | 1.2305 | 1.7146 | 0.9511 | 0.7400 | 0.8809 | 0.4766 |
| | 1.0 | 1.2282 | **1.7183** | 0.9596 | 0.7350 | **0.8808** | 0.4770 |
| | 1.1 | **1.2510** | 1.7117 | **0.9750** | **0.7432** | 0.8801 | **0.4811** |
| | 0.3 | 1.1484 | 1.6915 | 0.9005 | 0.7240 | 0.8809 | 0.4691 |
| 7 | 0.8 | 1.2218 | 1.7020 | 0.9405 | 0.7375 | 0.8811 | 0.4782 |
| | 1.1 | **1.2566** | **1.7062** | **0.9916** | **0.7422** | **0.8813** | **0.4834** |

Table 13: **Exponential decay factor** $\beta$ with all other settings fixed: `num_resampling=3`, `num_steps=5`, $\lambda = 0.9$. We report two blocks: `num_ts=6` and `num_ts=7`. (Best values for each block are bold.)

| num_ts | $\beta$ | ImageReward (↑) | | | BLIP-VQA (↑) | | |
|---|---|---|---|---|---|---|---|
| | | Animals | Animals_Objects | Objects | Animals | Animals_Objects | Objects |
| | 0.3 | 1.1437 | 1.7020 | 0.9518 | 0.7248 | 0.8816 | 0.4764 |
| | 0.7 | 1.2051 | 1.7130 | 0.9860 | 0.7407 | 0.8851 | 0.4788 |
| 6 | 0.8 | 1.2037 | 1.7110 | 0.9916 | 0.7388 | 0.8832 | **0.4894** |
| | 0.9 | 1.2003 | 1.7159 | 0.9883 | **0.7416** | **0.8842** | 0.4891 |
| | 1.1 | **1.2098** | **1.7180** | **0.9956** | 0.7371 | 0.8841 | 0.4855 |
| | 0.3 | 1.1273 | 1.6938 | 0.9490 | 0.7274 | 0.8848 | 0.4796 |
| 7 | 0.8 | 1.2142 | 1.7000 | 0.9937 | 0.7363 | 0.8845 | **0.4952** |
| | 1.1 | **1.2331** | **1.7046** | **1.0148** | **0.7496** | **0.8864** | 0.4945 |

### A.8.5 **CO3** FAILURE CASES

Despite correcting for the "problematic" modes, there still remain open challenges in image composition as shown in Figure 8. The score and, hence, the correction landscape is heavily influenced by the training schemes employed in diffusion model training, i.e., the quantity and quality of the multi-concept bindings in the training set. In addition, the usage of unrealistic prompts perhaps not encountered in training also results in poor text/concept alignment. We leave this investigation for the future.

### A.9 MORE QUALITATIVE RESULTS ON MULTI-CONCEPT PROMPTS

We added more qualitative comparison results on the three categories from Attend-Excite prompts in Figure 9, Complex category from T2Icompnench in Figure 10. Additional results on Rarebench prompts are included in Figure 11.

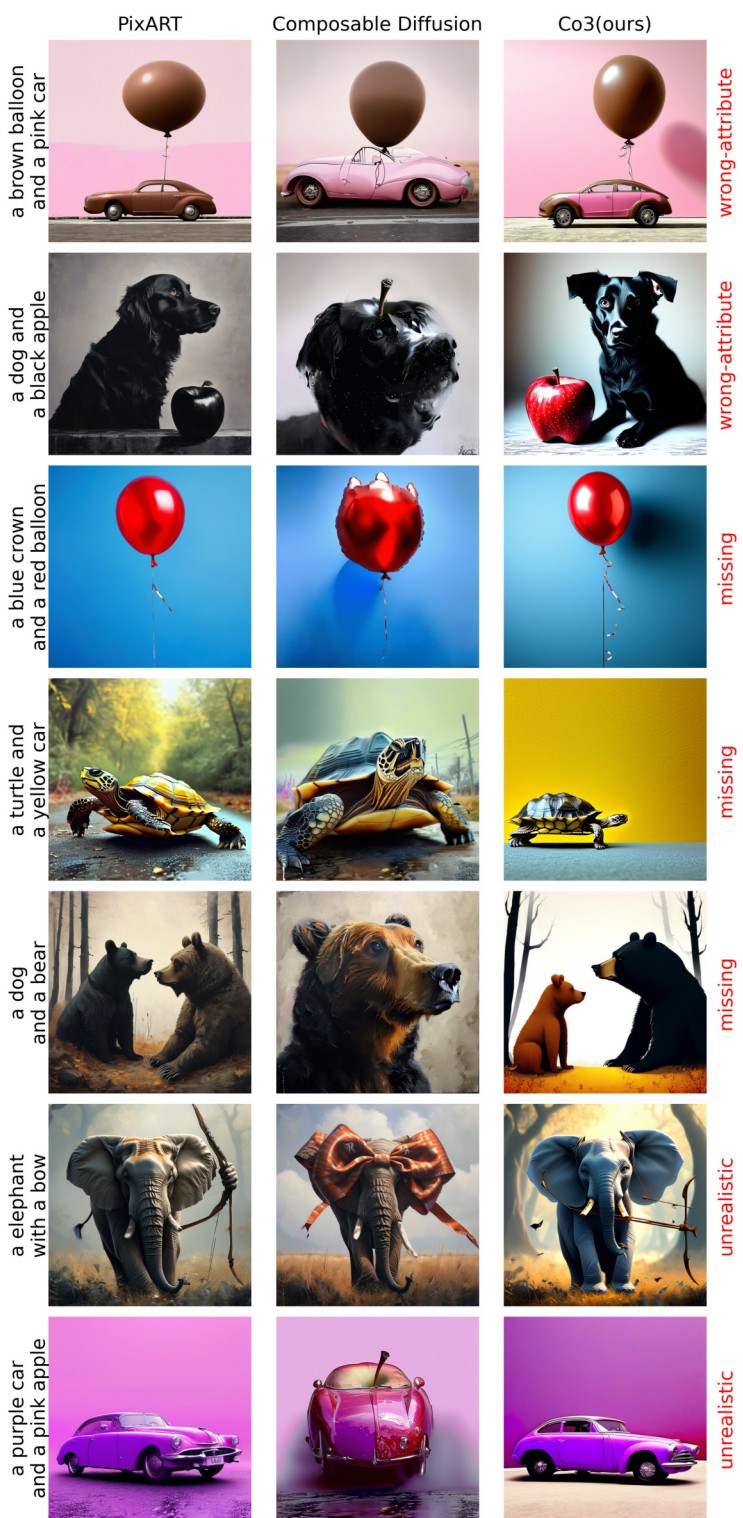

Figure 8: Failure scenarios of PixART-Σ base diffusion model, PixART-Σ + **CO3**, and PixART-Σ + Composable Diffusion.

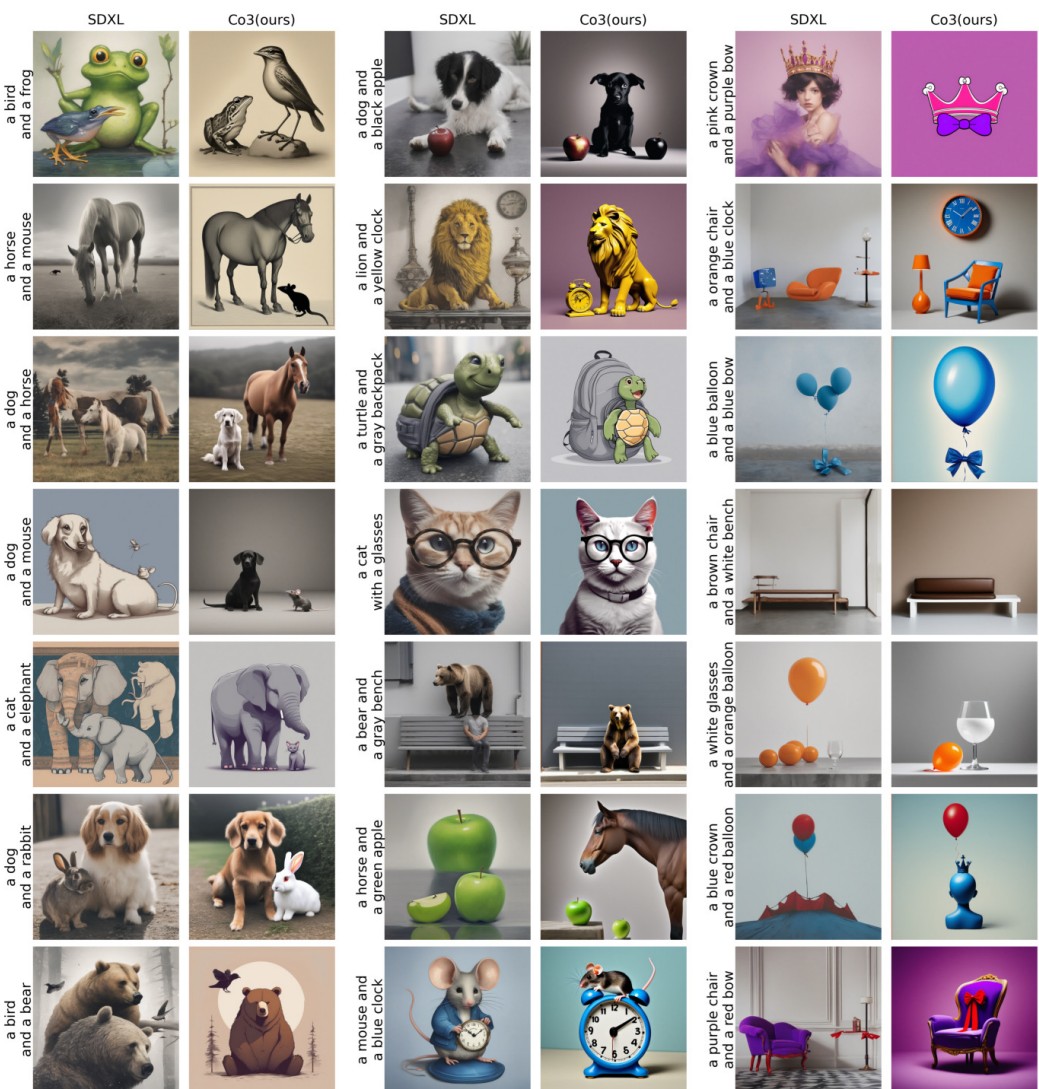

Figure 9: Qualitative results of SDXL base diffusion model, and SDXL + **CO3** for *animal-animal*,*animal-object*, and *object-object* categories from Attend-Excite prompts.

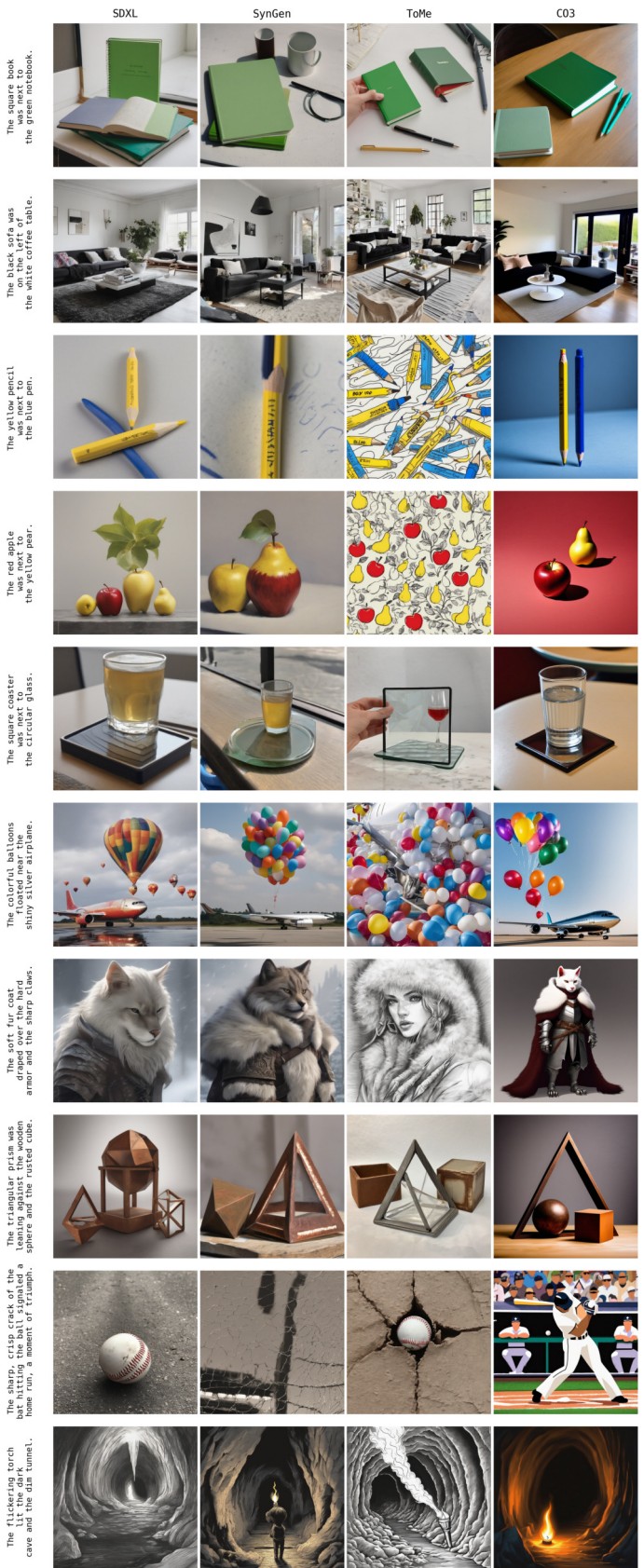

Figure 10: Qualitative results on Complex category from T2Icompbench

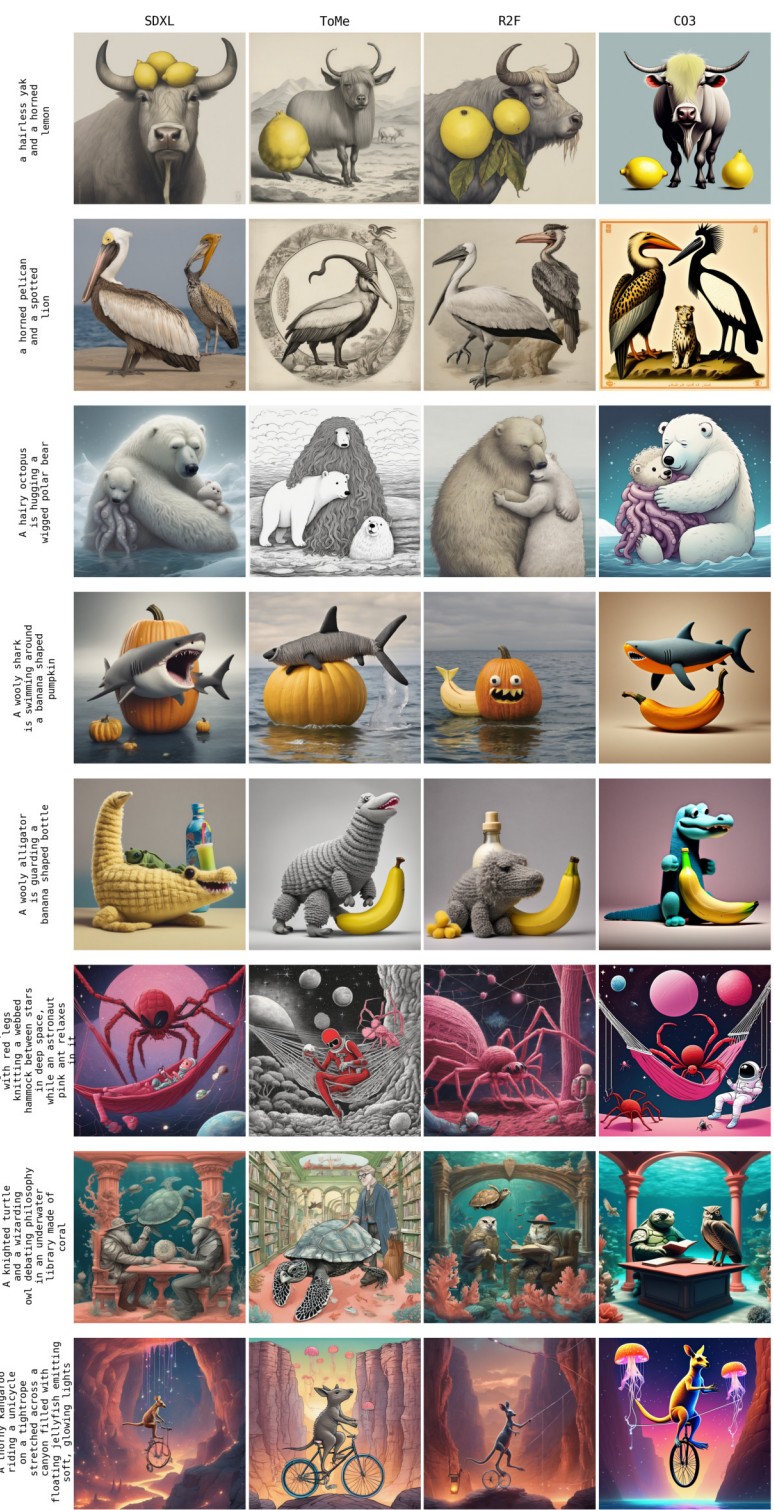

Figure 11: Qualitative results on RareBench prompts.

