# OpenReview forum: "Steer Away From Mode Collisions: Improving Composition In Diffusion Models"
_ICLR.cc/2026/Conference — ICLR 2026 Poster_

### Official Review · Reviewer_agn3 · 2025-10-27

**Soundness:** 3
**Presentation:** 3
**Contribution:** 3
**Rating:** 6
**Confidence:** 4

**Summary:**

This paper proposed the CO3, Contrasting concepts compose better, to improve muti-concept prompt fidelity in text-to-image diffusion models. Instead of re-training, the authors introduced a corrective sampling strategy, to steer away from regions where the joint prompt behavior overlaps strongly with any single concept in the prompt. Specifically, they analyze and show that composition through weighted sum of Tweedie-means in the Tweedie denoised space, offers a more general framework. Experiments on diverse multi-concept prompts demonstrate improvements in concept coverage, balance and robustness, with fewer dropped or distorted concepts compared to standard baselines and prior compositional works.

**Strengths:**

1)	The paper is well-written, and the structure is well-organized.
2)	The proposed CO3 is plug-and-play, model-agnostic, and gradient-free. And the results shows stronger semantic alignment to prompts.
3)	Combing the strengths of correction-based approaches and composable diffusion seem reasonable and effective.

**Weaknesses:**

1)	Some very latest and more relevant works are not compared, such as, Magnet (We Never Know How Text-to-Image Diffusion Models Work, Until We Learn How Vision-Language Models Function), TWEEDIEMIX ICLR 2025, ConceptWeaver, etc.
2)	It is confusing that why using arbitrary weights in (10) (i.e., Liu et al. 200) does not lead to a valid Tweedie-mean. There seems a lack of theoretical analysis or empirical validation.
3)	How to connect the equation (12) to (13). On other words, why (13) is representing the samping from the unnormalized probability distribution in Eq. (12).
4)	More experiments are tested on two concept prompts. Can this model generalize to the scenario with the prompts of three or more concepts?

**Questions:**

1)	Some very latest and more  closely relevant works are not compared, such as, Magnet (We Never Know How Text-to-Image Diffusion Models Work, Until We Learn How Vision-Language Models Function), TWEEDIEMIX ICLR 2025, ConceptWeaver, etc.
2)	It is confusing that why using arbitrary weights in (10) (i.e., Liu et al. 200) does not lead to a valid Tweedie-mean. There seems a lack of theoretical analysis or empirical validation.
3)	How to connect the equation (12) to (13). On other words, why (13) is representing the samping from the unnormalized probability distribution in Eq. (12).
4)	More experiments are tested on two concept prompts. Can this model generalize to the scenario with the prompts of three or more concepts?

---

> ### Author Response · Authors · 2025-11-26
>
> >__Q1. Some very latest and more relevant works are not compared, such as, Magnet (We Never Know How Text-to-Image Diffusion Models Work, Until We Learn How Vision-Language Models Function), TWEEDIEMIX ICLR 2025, ConceptWeaver, etc__
>
> Thank you for pointing out these recent work. We have now added new results on Magnet and TweedieMix. We note that both TweedieMix and ConceptWeaver operate in the concept-customization regime: they require a fine-tuned LoRA (or customized embedding) for each concept involved. A fair, apples-to-apples comparison on our benchmarks and baselines would therefore require (i) collecting or generating additional training images for every concept, and (ii) fine-tuning a separate LoRA model for each of the hundreds of concepts that appear across our prompts - both computationally infeasible and fundamentally different from our setting. CO3, in contrast, is a training-free, model-agnostic inference method that does not rely on concept-specific customization. To still provide insight, we adapt TweedieMix’s sampling strategy only (without its customization stage) and report these results below.  We see that, CO3 still outperforms both Tweediemix and Magnet almost across all categories (except for Magnet’s strong performance in Objects-Objects in BLIP-VQA metric).
>
> &nbsp;
>
> **Legend: IR: ImageReward, VQA: BLIP-VQA
>    &nbsp;&nbsp;&nbsp;&nbsp;&nbsp;&nbsp;&nbsp;&nbsp;&nbsp;&nbsp;&nbsp;&nbsp; &nbsp;&nbsp;        A-A: animals-animals, A-O: animals-objects, O-O: objects-objects**
>
> | Methods     | IR(A-A) $\uparrow$  | IR(A-O) $\uparrow$ | IR(O-O) $\uparrow$ | VQA(A-A) $\uparrow$ | VQA(A-O) $\uparrow$ | VQA(O-O) $\uparrow$ |
> |-------------|----------|----------|----------|-----------|-----------|-----------|
> | SDXL        | 0.4906   | 1.1503   | 0.1131   | 0.6428    | 0.8378    | 0.4752    |
> | TweedieMix  | 1.0023   | 1.3127   | 0.7959   | 0.7390    | 0.8059    | 0.4683    |
> | Magnet      | 0.6450   | 1.5895   | 0.7856   | 0.6782    | 0.8744    | 0.5398    |
> | ToMe        | 0.3894   | 1.5736   | 1.0117   | 0.6257    | 0.8807    | 0.6440    |
> | Co3         | 1.2341   | 1.6743   | 1.0158   | 0.7440    | 0.8877    | 0.5145    |

---

> ### Author Response · Authors · 2025-11-26
>
> >__Q2. It is confusing that why using arbitrary weights in (10) (i.e., Liu et al. 200) does not lead to a valid Tweedie-mean. There seems a lack of theoretical analysis or empirical validation.__
>
> Thank you for raising this question and we apologize if our explanation was not clear.
>
> In our setting, for a fixed linear–Gaussian forward process, the Tweedie mean at step $t$ under classifier-free guidance (CFG) has the form
>
> $\hat{x}\_{tweedie}(x\_t, c)
> = (x\_t - \sqrt{1-\bar{\alpha}\_t}\,\epsilon\_\theta^{\mathrm{cfg}}(x\_t, c, t)) / \sqrt{\bar{\alpha}\_t}.$
>
>
>
> This can be rewritten as
>
> $$
> \hat{x}_0(x_t, c) = a_t x_t + b_t(c),
> $$
>
> where $a\_t = 1 / \sqrt{\bar{\alpha}\_t}$
>  is fixed by the noise schedule and does not depend on the conditioning $c$. The term $\epsilon_\theta^{cfg}\$ is the usual linear combination of conditional and unconditional noise predictions.
>
> Now consider the linear combination of scores used in Eq. (11) of Liu et al.[1]. If we use it in the expression at the top, for $\tilde{x}_{tweedie}$ to be the Tweedie mean of some conditional joint distribution $p(x_t \mid c_1, c_2,..,c_k)$ under the same forward process, its dependence on $x_t$ must again take the form $a\_t x\_t + b\_t(\cdot)$ with the same coefficient $a\_t$. This forces the normalization constraint
>
> $$
> \sum_i w_i = 1.
> $$
>
> Thus, when mixing scores from different conditional distributions, the above condition need to be satisfied for $\tilde{x}\_{tweedie}$ to correspond to any coherent joint conditional model. Eq. (11) of Liu et al. allows arbitrary weights, which in general violate these constraints. In such cases, the result is simply a heuristic linear combination of scores, and the resultant Tweedie mean doesn't correspond to any well-defined distribution under the given forward process. This is what we meant by “does not lead to a valid Tweedie mean.”
>
> By contrast, CO3 composes Tweedie-means from different conditional distributions. Lemma 1 shows that these weights have to satisfy the same unit-sum condition as above for the resultant composed Tweedie-mean to be a valid one.  CO3 maintains this condition and  this preserves the update on the Tweedie manifold of the underlying sampler.
>
> We will augment the derivation in Appendix A.4 with this explanation.
>
> &nbsp;
>
> [1] Liu, Nan, Shuang Li, Yilun Du, Antonio Torralba, and Joshua B. Tenenbaum. "Compositional visual generation with composable diffusion models." In European conference on computer vision, pp. 423-439. Cham: Springer Nature Switzerland, 2022.

---

> ### Author Response · Authors · 2025-11-26
>
> >__Q3. How to connect the equation (12) to (13). On other words, why (13) is representing the sampling from the unnormalized probability distribution in Eq. (12).__
>
> &nbsp;
>
> Thank you for raising this question.
>
> Eq. (12) defines the CO3 corrector distribution at timestep $t$ by down-weighting regions where the joint distribution $p_t(x \mid C) $ overlaps with individual concept distributions $p_t(x \mid c_i) $. It introduces the unnormalized density:
>
> $$
> \log \tilde{p}\_t(x \mid C)
> = w\_0 \log p\_t(x \mid C) - \sum\_{i=1}^K w\_i \log p\_t(x \mid c\_i).
> $$
>
> The resulting score is a weighted linear combination of per-prompt scores.
>
> By Tweedie’s formula, for any conditioning $c$, the Tweedie mean satisfies:
>
> $$
> \hat{x}\_{tweedie}(x\_t, c)
> = x\_t + \sigma\_t \nabla\_{x\_t} \log p\_t(x\_t \mid c),
> $$
>
> where $\sigma_t$ depends only on the noise schedule. Therefore, any linear combination of scores corresponds to the same linear combination of Tweedie means (up to the shared $x_t$ term).
>
> Eq. (13) implements this by composing the joint and per-concept Tweedie means with the same weights. Plugging this into the DDIM update yields a probability-flow sampler (Eq. 5–6), and iterating at fixed $t$ samples from $\tilde{p}_t(x \mid C)$ (see Algorithms 2 & 3).
>
> Finally, we emphasize that composing in Tweedie-mean space is more general than working directly in score space. As proven in the paper (Lemma 1), any sampler whose update can be expressed in terms of $\hat{x}\_{tweedie}$ can adopt the same CO3 composition rule. This provides a unified view that recovers standard compositional-score methods as a special case.

---

> ### Author Response · Authors · 2025-11-26
>
> >__Q4. More experiments are tested on two concept prompts. Can this model generalize to the scenario with the prompts of three or more concepts?__
>
> Yes, CO3 generalizes to 3 or more concepts. In addition to our main evaluations on Attend-Excite prompts (which contain relatively simple two-concept compositions), we have included new experiments on the Complex category of T2I-CompBench as well as the Concat, Relation, and Complex categories of RareBench. These subsets were chosen because they specifically stress multi-concept interactions, attribute entanglement, and rare or low-support combinations. Compared to the simpler Attend-Excite cases, these categories contain more challenging, high-entropy compositions and therefore serve as stronger tests of compositional fidelity.  As pointed out by R2F [1],  BLIP-VQA can be unreliable in the rare-concepts regime. We follow the RareBench recommendation and report metrics that better reflect human judgement. Specifically, we report ImageReward and human evaluation from 11 participants.
>
> &nbsp;
> On T2ICompbench complex category:
>
> | Models    | ImageReward $\uparrow$     | BLIP-VQA  $\uparrow$ | Human Eval  $\uparrow$ |
> |-----------|--------|----------|--------|
> | SDXL &nbsp; &nbsp; &nbsp; &nbsp; &nbsp; &nbsp;     | 0.3083 &nbsp; &nbsp;  | 0.4170   | 0.5983 |
> | SynGen    | 0.2889 | 0.4083   | 0.5133 |
> | ToMe      | 0.2891 | 0.4544   | 0.5866 |
> | R2F       | 0.3561 | 0.4423   | 0.4816 |
> | Co3       | 0.4406 | 0.4761   | 0.8033 |
>
> &nbsp;
>
> On RareBench prompts with categories Concat, Relations and Complex:
> &nbsp;
>
> | Method | ImageReward (Concat)  $\uparrow$ | ImageReward (Relations)  $\uparrow$| ImageReward (Complex) $\uparrow$ | Human Eval (Concat)  $\uparrow$| Human Eval (Relations) $\uparrow$ | Human Eval (Complex) $\uparrow$ |
> |--------|----------------------|-------------------------|------------------------|----------------|-------------------|-----------------|
> | SDXL   | 0.1018 | 0.3607 | 1.1907 | 0.5416 | 0.5016 | 0.6283 |
> | ToMe   | -0.2041 | -0.0632 | 1.0200 | 0.4500 | 0.4983 | 0.48666 |
> | R2F    | -0.0162 | 0.3620 | 1.1542 | 0.4300 | 0.5200 | 0.53666 |
> | Co3    | 0.1610 | 0.2588 | 1.2211 | 0.6167 | 0.6200 | 0.61166 |
>
> &nbsp;
>
> [1] Park, Dongmin, Sebin Kim, Taehong Moon, Minkyu Kim, Kangwook Lee, and Jaewoong Cho. "Rare-to-frequent: Unlocking compositional generation power of diffusion models on rare concepts with LLM guidance." arXiv preprint arXiv:2410.22376 (2024)

---

### Official Review · Reviewer_Md3g · 2025-10-31

**Soundness:** 2
**Presentation:** 3
**Contribution:** 2
**Rating:** 4
**Confidence:** 4

**Summary:**

The authors proposed a method for better prompt-aligned multi-concept image generation. Specifically, existing models often neglect or mix multiple concepts in a single generation scenario. The proposed method tends to squeeze the distribution, hindering the generation process from walking through some problematic regions. The probability normalization approach is designed for this purpose.

**Strengths:**

1. The diagnosis of the multi-concept image generation problem is interesting—specifically, the overlap of individual concept distributions and the original multi-concept distribution's regions.
2. The distribution correction idea is theoretically intuitive and good.

**Weaknesses:**

1. A known post-hoc guidance is called in a new name, “correction guidance”, for a better sound of the paper. It sounds unnatural.
2. Only Subjective metrics were used. What about objective ones, such as CLIP-T, DSG (Davidsonian Scene Graph), or simple classification metrics?
3. The proposed method is based on a modern and relatively stronger model, SDXL, while it is often compared with older methods with weaker diffusion models (2023 methods, such as Attend and Excite or Divide-Bind with SD 1.5 or 2.0). Also, many other modern methods are neglected in the paper ( Not a fair comparison.
4. In the ablation study, the corrector’s contribution is somehow questioned: a) What is the performance of SDXL+Corrector (w/o Re-sampler)? b) In all 6 metrics, it has gains only in 3 metrics, out of 6. The other 3 are either the same or even worse.
5. There is no limitation section.
6. In Table 3, it has the worst speed performance, and even the metrics are not great compared with ToMe.
7. Limited color diversity in qualitative results in Figure 3.
8. No qualitative Ablation study.
9. It drops the image quality in Figure 5.

**Questions:**

1. Isn’t \epsilon have a similar formulation as \tilda(\epsilon) in Lemma 1 204-205 (CFG, with Lambda)? Is it a typo?
2. What is the "x" in Lemma 1, a) 209? is it x_{t}?

---

> ### Author Response · Authors · 2025-11-25
>
> >__Q1. The proposed method is based on a modern and relatively stronger model, SDXL, while it is often compared with older methods with weaker diffusion models (2023 methods, such as Attend and Excite or Divide-Bind with SD 1.5 or 2.0). Also, many other modern methods are neglected in the paper.__
>
>
> We understand the concern. To support the claim of generality beyond SDXL, we have now added evaluations on an additional open-source text-to-image model (PixArt-$\Sigma$). On this backbone, CO3 improves over both the base PixArt model and its model-agnostic composable-diffusion baseline, demonstrating that the method transfers across architectures without requiring any re-training or model-specific modifications.
>
> In addition, we have also added more recent baselines, namely Magnet [1] and Tweediemix [2]  (suggested by reviewer agn3), for evaluation on A&E prompts. Co3 remains the best-performing model on SDXL demonstrating the effectiveness of the proposed method in compositional generation.
> Visualizations for these experiments are added in Figure 4 & 5 of the main text.
>
> &nbsp;
>
> **Legend: IR: ImageReward, VQA: BLIP-VQA
> &nbsp;&nbsp;&nbsp;&nbsp;&nbsp;&nbsp;&nbsp;&nbsp;&nbsp;&nbsp; A-A: animals-animals, A-O: animals-objects, O-O: objects-objects**
> | Method | IR (A-A)$\uparrow$ | IR (A-O)$\uparrow$ | IR (O-O)$\uparrow$ | VQA (A-A)$\uparrow$ | VQA (A-O)$\uparrow$ | VQA (O-O)$\uparrow$ |
> |--------|----------|----------|----------|-----------|-----------|-----------|
> | Co3 &nbsp;&nbsp;&nbsp;&nbsp;| 1.2341 | 1.67431 | 1.0158 | 0.7440 | 0.8877 | 0.5145 |
> | Magnet | 0.645005 | 1.589587 | 0.7856 | 0.6782 | 0.8744 | 0.5398 |
> | Tweediemix | 1.0023 | 1.3127 | 0.7959 | 0.7390 | 0.8059 | 0.4683 |
> | PiXArt | 1.0664 | 1.6973 | 1.0237 | 0.6912 | 0.8615 | 0.5598 |
> | PixArt + composableDiffusion | 0.9170 | 0.0479 | 0.3391 | 0.208 | 0.5914 | 0.4721 |
> | PixArt + Co3 | 1.3805 | 1.7704 | 1.0811 | 0.7463 | 0.8644 | 0.5815 |
> &nbsp;
>
> [1] Zhuang, Chenyi, Ying Hu, and Pan Gao. "Magnet: We never know how text-to-image diffusion models work, until we learn how vision-language models function." Advances in Neural Information Processing Systems 37 (2024): 57115-57149.
> [2] Kwon, Gihyun, and Jong Chul Ye. "Tweediemix: Improving multi-concept fusion for diffusion-based image/video generation." arXiv preprint arXiv:2410.05591 (2024).

---

> ### Author Response · Authors · 2025-11-25
>
> >__Q2a. In the ablation study, the corrector’s contribution is somehow questioned: a) What is the performance of SDXL+Corrector (w/o Re-sampler)?__
>
> We update the ablation table with the additional entry of SDXL+Corrector.
>
> Including the Corrector alone already improves performance in categories where attributes must attach to specific objects. This is most clearly visible in Rows 4–5 of the ablation table: prompts of the form “a [attribute1][object1] and [attribute2][object2]” show large gains, because early-timestep dominance (one object absorbing both attributes) is the failure mode that the Corrector resolves.
>
>
> &nbsp;
>
> **Legend: IR: ImageReward, VQA: BLIP-VQA
> &nbsp;&nbsp;&nbsp;&nbsp;&nbsp;&nbsp;&nbsp;&nbsp;&nbsp;           A-A: animals-animals, A-O: animals-objects, O-O: objects-objects**
>
> | Method | IR (A-A)$\uparrow$ | IR (A-O)$\uparrow$ | IR (O-O)$\uparrow$ | VQA (A-A)$\uparrow$ | VQA (A-O)$\uparrow$ | VQA (O-O) $\uparrow$ | Avg$\uparrow$ |
> |--------|----------|----------|----------|-----------|-----------|-----------|-----|
> | SDXL | 0.7820 | 1.5474 | 0.6789 | 0.6951 | 0.8654 | 0.4926 | 0.8435 |
> | SDXL + Resampling | 1.0881 | 1.6755 | 0.8263 | 0.7351 | 0.8666 | 0.4528 | 0.9437 |
> | SDXL + Resampling + corrector | 1.0630 | 1.6429 | 0.8949 | 0.7177 | 0.8794 | 0.4796 | 0.9463 |
> | SDXL + Resampling + Corrector + weight-modulation | 1.2342 | 1.6744 | 1.0158 | 0.7441 | 0.8878 | 0.5146 | 1.0118 |
> | SDXL + Corrector | 0.9543 | 1.5140 | 0.8894 | 0.7149 | 0.8801 | 0.4989 | 0.9086 |
> | SDXL + Corrector + weight-modulation | 1.1732 | 1.5865 | 1.0067 | 0.7211 | 0.8701 | 0.5206 | 0.9797 |
>
> &nbsp;
>
> >__Q2.b In all 6 metrics, it has gains only in 3 metrics, out of 6. The other 3 are either the same or even worse__
>
> The Corrector specifically targets concept–attribute misalignment, so it is understandable that its performance is not pronounced in categories such as animals-animals (i.e., prompts of the form *“a [animal1] and a [animal2]”*). Thus, improvements appear strongest in the categories that reflect attribute fidelity and multi-object separation.  Importantly, the average performance of *SDXL + Corrector + weight-modulation (last row)* is very close to the *SDXL + Resampling + Corrector + weight-modulation (4th row)*  which indicates the Corrector is powerful even without Resampling. But the full improvement is achieved when the Corrector is applied after the Resampling steps, so that weight-modulation together with the Corrector mitigates concept dominance and improves concept-attribute alignment.

---

> ### Author Response · Authors · 2025-11-26
>
> >__Q3. In Table 3, it has the worst speed performance, and even the metrics are not great compared with ToMe__
>
> Although CO3 takes more time compared to ToMe with 50 time-steps, it takes much less memory. CO3 is light-weight and it’s peak memory requirement is only slightly above the base SDXL model. ToMe is gradient based methods and has significant memory requirement.
> Also we want to emphasize that CO3 is model-agnostic and can be applied to any diffusion-based model. Compared to that, ToMe is model-architecture dependent where its attribute binding approach works by manipulating the internal attention activations of the base-model.
>
> For better comparison, we also tested CO3 with 20 inference steps and the Table below shows that the average performance of CO3 with 20 steps is still faster/better than ToMe with 50 steps. CO3 with 20 steps takes 10.65 secs which is less than ToMe.
>
> &nbsp;
>
> | Method | IR (A-A)$\uparrow$ | IR (A-O)$\uparrow$ | IR (O-O)$\uparrow$ | VQA (A-A)$\uparrow$ | VQA (A-O)$\uparrow$ | VQA (O-O)$\uparrow$ | Avg$\uparrow$ | Time$\downarrow$ | Memory$\downarrow$ |
> |--------|----------|----------|----------|-----------|-----------|-----------|-----|------|--------|
> | SDXL-50 steps  | 0.4906 | 1.1503 | 0.1131 | 0.6428 | 0.8378 | 0.4752 | 0.6183 | 7.20 | 8.5GB |
> | ToMe-50 steps | 0.3894 | 1.5736 | 1.0117 | 0.6257 | 0.8807 | 0.6440 | 0.8542 | 16.58 | 18 GB |
> | CO3-50 steps | 1.2341 | 1.6743 | 1.0158 | 0.7440 | 0.8877 | 0.5145 | 1.012 | 19.9 | 9.5 GB |
> | CO3-20 steps | 0.8532 | 1.6440 | 0.7481 | 0.6698 | 0.8611 | 0.4315 | 0.8679 | 10.65 | 9.5 GB |
>
> &nbsp;
>
> >__Q4. No qualitative Ablation study.__
>
> We have added new figures to Appendix A.7.9.
>
> &nbsp;
>
> >__Q5. There is no limitation section.__
>
> We apologize for not including the limitation section before. We are updating the paper with such as section, mainly discussing the following three items:
> - In obeying the prompt, CO3 seems to be suppressing the richness of the background. We would like to offer control on background generation without sacrificing the prompt obedience.
> - CO3 lacks a mechanism to adaptively select/adjust the number of Resampling or Corrector steps.
> - Similar to other baselines, CO3 has room for improvement in concept-attribute alignment.
>
> &nbsp;
>
> >__Q6. Low  image quality in Figure 5. Limited color diversity in qualitative results in Figure 3.__
>
> Thanks for raising an interesting observation. We investigated it further. What initially appears as reduced visual richness in a few simple cases (e.g., Fig. 5) is in fact a consistent manifestation of *CO3’s stronger prompt obedience*. CO3 tends to strongly obey the concepts specified in the prompt; when the background is not specified it is conservative with hallucinating the background. However if the background is explicitly specified, CO3 brings the same degree of richness and quality to both the foreground and the background.
> We have added a discussion and new figures in Appendix A.7.8, that enriches the "low quality" images in Figure 5 based on explicitly specified backgrounds.
>
> &nbsp;
>
> >__Q7. Isn’t $\epsilon$ have a similar formulation as a)  Isn’t $\epsilon$ have a similar formulation as $\tilde{\epsilon}$ in Lemma 1 204-205 (CFG, with Lambda)? Is it a typo? b) What is the "x" in Lemma 1, a) 209? is it x_t__
>
> We thank the reviewer for reading the paper in such detail. Yes, both were typos. **(a)** We have updated the main paper with $\tilde{\epsilon}$ everywhere.  **(b)** We have changed $x$ to $x_t$ in the paper.

---

> ### Author Response · Authors · 2025-11-26
>
> >__Q8. Only Subjective metrics were used. What about objective ones, such as CLIP-T, DSG (Davidsonian Scene Graph), or simple classification metrics__
>
> Thanks for raising this point.
>
> We agree that more metrics will validate the robustness of the proposed method. For this, we include  DINOv2 [1] score as a classification metric for the concepts.  This score will compare the quality of concepts generated by each model.
>
> For each prompt, we extract the concepts from the prompt text and query the pretrained DINOv2 model to output a score for each concept in the generated image. We report the mean concept score across all prompts in the Table below. We evaluate on our most competitive SDXL-based baselines from Table 1 on A&E prompts and also have added results on two new baselines, namely  Magnet [2] and Tweediemix [3]. CO3 improves upon the baselines on this metric also, demonstrating higher quality concept generation capability.
>
> &nbsp;
>
> **Legend: DINO: DINOv2 score
> &nbsp; &nbsp;&nbsp; &nbsp;&nbsp; &nbsp;&nbsp; &nbsp;&nbsp; A-A: animals-animals, A-O: animals-objects, O-O: objects-objects**
> | Method | DINO (A-A)$\uparrow$ | DINO (A-O)$\uparrow$ | DINO (O-O)$\uparrow$ |
> |--------|------------|------------|------------|
> | SDXL | 0.2234 | 0.2618 | 0.2629 |
> | SynGen | 0.2221 | 0.2650 | 0.2710 |
> | ToMe | 0.2226 | 0.2620 | 0.2545 |
> | Magnet | 0.2122 | 0.2636 | 0.2624 |
> | Tweediemix | 0.2519 | 0.3105 | 0.3237 |
> | CO3 | 0.2623 | 0.3259 | 0.3283 |
>
> &nbsp;
>
> [1] Oquab, Maxime, Timothée Darcet, Théo Moutakanni, Huy Vo, Marc Szafraniec, Vasil Khalidov, Pierre Fernandez et al. "Dinov2: Learning robust visual features without supervision." arXiv preprint arXiv:2304.07193 (2023).
> [2] Zhuang, Chenyi, Ying Hu, and Pan Gao. "Magnet: We never know how text-to-image diffusion models work, until we learn how vision-language models function." Advances in Neural Information Processing Systems 37 (2024): 57115-57149.
> [3] Kwon, Gihyun, and Jong Chul Ye. "Tweediemix: Improving multi-concept fusion for diffusion-based image/video generation." arXiv preprint arXiv:2410.05591 (2024).

---

### Official Review · Reviewer_3fep · 2025-10-31

**Soundness:** 2
**Presentation:** 3
**Contribution:** 3
**Rating:** 4
**Confidence:** 4

**Summary:**

This paper tackles the problem of multi-concept compositional generation in text-to-image diffusion models. The authors hypothesize such models struggle with compositional generation because the learned joint distribution for composite prompts overlaps significantly with individual concept distributions, leading to dominance by a single concept. To address this, they propose CO3, a plug-and-play corrective sampling method that suppresses individual concepts by re-weighting Tweedie means of each concept, rather than directly re-weighting in the noise (or score) space. Experiments on SD1.5, SDXL, and Pixart-$\Sigma$ show improved prompt alignment and concept coexistence.

**Strengths:**

1. CO3 requires no retraining or gradient computation and can be applied to any diffusion model. This makes it broadly applicable and practical.
2. The paper provides an intuitive theoretical explanation showing that directly performing weighted averaging of scores does not guarantee a valid Tweedie mean or consistent classifier-free guidance formulation; by instead re-weighting in the Tweedie-denoised space, CO3 ensures mathematical validity.
3. Novelty. While not the first to explore Tweedie-denoised space reweighting for concept composition [1], CO3 offers a clear theoretical formulation and a practical two-stage approach, early resampling followed by later correction, that together provide consistent empirical improvements and strong interpretability.

[1] Kwon & Ye, TweedieMix: Improving Multi-Concept Fusion for Diffusion-based Image/Video Generation, ICLR 2025.

**Weaknesses:**

My main concern lies in the evaluation, particularly regarding benchmark coverage and baseline selection.

1. The evaluated prompts are relatively limited, focusing mainly on compositions involving animals and objects. Incorporating commonly used benchmarks such as T2I-CompBench, which include more diverse attribute and object combinations, would strengthen the empirical validation.
2. A relevant baseline is missing [1]. In addition, comparisons with recent **LLM-guided generation methods** [2, 3, 4] would be beneficial to quantitatively assess the relative performance and effectiveness of CO3 in enhancing compositional fidelity. If the proposed method is orthogonal to LLM-guided approaches, it would be particularly valuable to demonstrate how CO3 complements them when combined.
3. Including comparisons with recent **state-of-the-art text-to-image models**, such as **Stable Diffusion 3.5** or **Flux**, would improve the paper’s relevance and help readers better understand the practical significance of the proposed method.

[1] Yu et al., Improving Compositional Generation with Diffusion Models Using Lift Scores, ICML 2025.

[2] Yang et al., Mastering text-to-image diffusion: Recaptioning, planning, and generating with multimodal llms, ICML 2024.

[3] Lian et al., LLM-grounded Diffusion: Enhancing Prompt Understanding of Text-to-Image Diffusion Models with Large Language Models, TMLR 2024.

[4] Hu et al., ELLA: Equip Diffusion Models with LLM for Enhanced Semantic Alignment, arXiv preprint.

**Questions:**

Please refer to weaknesses section.

---

> ### Author Response · Authors · 2025-11-25
>
> >__Q1. The evaluated prompts are relatively limited, focusing mainly on compositions involving animals and objects. Incorporating commonly used benchmarks such as T2I-CompBench, which include more diverse attribute and object combinations, would strengthen the empirical validation.__
>
> We appreciate the reviewer’s suggestion regarding broader benchmark coverage. In addition to our main evaluations on Attend-Excite prompts (which contain relatively simple two-concept compositions), we have included new experiments on the Complex category of T2I-CompBench as well as the Concat, Relation, and Complex categories of RareBench. These subsets were chosen because they specifically stress multi-concept interactions, attribute entanglement, and rare or low-support combinations, which align with the failure modes CO3 is designed to address. Compared to the simpler Attend-Excite cases, these categories contain more challenging, high-entropy compositions and therefore serve as stronger tests of compositional fidelity.  As pointed out by R2F [1],  BLIP-VQA can be unreliable in the rare-concepts with complex relationship regime, hence we follow the RareBench recommendation and report metrics that better reflect human judgement. Specifically, we report ImageReward and human evaluation conducted with 11 participants following the protocols from [1].
>
> &nbsp;
> On T2ICompbench complex category:
>
> | Models    | ImageReward $\uparrow$     | BLIP-VQA  $\uparrow$ | Human Eval  $\uparrow$ |
> |-----------|--------|----------|--------|
> | SDXL &nbsp; &nbsp; &nbsp; &nbsp; &nbsp; &nbsp;     | 0.3083 &nbsp; &nbsp;  | 0.4170   | 0.5983 |
> | SynGen    | 0.2889 | 0.4083   | 0.5133 |
> | ToMe      | 0.2891 | 0.4544   | 0.5866 |
> | R2F       | 0.3561 | 0.4423   | 0.4816 |
> | Co3       | 0.4406 | 0.4761   | 0.8033 |
>
> &nbsp;
>
> On RareBench prompts with categories Concat, Relations and Complex:
> &nbsp;
>
> | Method | ImageReward (Concat)  $\uparrow$ | ImageReward (Relations)  $\uparrow$| ImageReward (Complex) $\uparrow$ | Human Eval (Concat)  $\uparrow$| Human Eval (Relations) $\uparrow$ | Human Eval (Complex) $\uparrow$ |
> |--------|----------------------|-------------------------|------------------------|----------------|-------------------|-----------------|
> | SDXL   | 0.1018 | 0.3607 | 1.1907 | 0.5416 | 0.5016 | 0.6283 |
> | ToMe   | -0.2041 | -0.0632 | 1.0200 | 0.4500 | 0.4983 | 0.48666 |
> | R2F    | -0.0162 | 0.3620 | 1.1542 | 0.4300 | 0.5200 | 0.53666 |
> | Co3    | 0.1610 | 0.2588 | 1.2211 | 0.6167 | 0.6200 | 0.61166 |
>
> &nbsp;
>
> [1] Park, Dongmin, Sebin Kim, Taehong Moon, Minkyu Kim, Kangwook Lee, and Jaewoong Cho. "Rare-to-frequent: Unlocking compositional generation power of diffusion models on rare concepts with LLM guidance." arXiv preprint arXiv:2410.22376 (2024).

---

> ### Author Response · Authors · 2025-11-25
>
> >__Q2.Comparison with recent LLM-based generation methods__
>
> We have compared against R2F [1] (ICLR 2025) which outperforms the three LLM based methods (RPG [2], LMD[3], ELLA [4] ) you suggested. Below are the comparisons results against R2F:
>
> &nbsp;
>
> On T2ICompbench Complex category:
> &nbsp;
>
> | Models    | ImageReward $\uparrow$     | BLIP-VQA  $\uparrow$ | Human Eval  $\uparrow$ |
> |-----------|--------|----------|--------|
> | R2F       | 0.3561 | 0.4423   | 0.4816 |
> | Co3       | 0.4406 | 0.4761   | 0.8033 |
>
> &nbsp;
>
> On RareBench's Concat, Relations and Complex categories:
>
>
> | Method | ImageReward (Concat)  $\uparrow$ | ImageReward (Relations)  $\uparrow$| ImageReward (Complex) $\uparrow$ | Human Eval (Concat)  $\uparrow$| Human Eval (Relations) $\uparrow$ | Human Eval (Complex) $\uparrow$ |
> |--------|----------------------|-------------------------|------------------------|----------------|-------------------|-----------------|
> | R2F    | -0.0162 | 0.3620 | 1.1542 | 0.4300 | 0.5200 | 0.53666 |
> | Co3    | 0.1610 | 0.2588 | 1.2211 | 0.6167 | 0.6200 | 0.61166 |
>
> &nbsp;
>
> [1] Park, Dongmin, Sebin Kim, Taehong Moon, Minkyu Kim, Kangwook Lee, and Jaewoong Cho. "Rare-to-frequent: Unlocking compositional generation power of diffusion models on rare concepts with LLM guidance." arXiv preprint arXiv:2410.22376 (2024).
> [2] Yang et al., Mastering text-to-image diffusion: Recaptioning, planning, and generating with multimodal llms, ICML 2024.
> [3] Lian et al., LLM-grounded Diffusion: Enhancing Prompt Understanding of Text-to-Image Diffusion Models with Large Language Models, TMLR 2024.
> [4] Hu et al., ELLA: Equip Diffusion Models with LLM for Enhanced Semantic Alignment, arXiv preprint

---

> ### Author Response · Authors · 2025-11-25
>
> >__Q3. Including comparisons with recent state-of-the-art text-to-image models, such as Stable Diffusion 3.5 or Flux, would improve the paper’s relevance and help readers better understand the practical significance of the proposed method.__
>
> Thank you for the suggestion. To support the claim of generality beyond SDXL, we have now added evaluations on an additional open-source text-to-image model (PixArt-$\Sigma$). We compare only between model-agnostic methods, namely CO3 and composable-Diffusion. On this backbone, CO3 improves over both the base PixArt-$\Sigma$ model and the composable-Diffusion baseline on Attend-Excite prompts, demonstrating that the method transfers across architectures without requiring any re-training or model-specific modifications. Visualizations for these experiments were presented in Figure 5 of the main text.
> &nbsp;
>
> **Legend: IR: ImageReward, VQA: BLIP-VQA
> &nbsp;&nbsp;&nbsp;&nbsp;&nbsp;&nbsp;&nbsp;&nbsp; &nbsp;&nbsp;&nbsp;&nbsp;&nbsp;&nbsp; A-A: animals-animals, A-O: animals-objects, O-O: objects-objects**
> &nbsp;
> | Method | IR (A-A)$\uparrow$ | IR (A-O)$\uparrow$ | IR (O-O)$\uparrow$ | VQA (A-A)$\uparrow$ | VQA (A-O)$\uparrow$ | VQA (O-O)$\uparrow$ |
> |--------|----------|----------|----------|-----------|-----------|-----------|
> | PiXArt | 1.0664 | 1.6973 | 1.0237 | 0.6912 | 0.8615 | 0.5598 |
> | PixArt + composable-Diffusion | 0.917 | 0.0479 | 0.3391 | 0.208 | 0.5914 | 0.4721 |
> | PixArt + Co3 | 1.3805 | 1.7704 | 1.0811 | 0.7463 | 0.8644 | 0.5815 |
>
> &nbsp;
>
> In the short rebuttal period, we were able to extend to PixArt-$\Sigma$ but not to the FLUX backbone. FLUX uses a hybrid architecture and currently lacks a complete technical/scientific report, so integrating CO3 with it would require careful testing to ensure faithful and reproducible results. To avoid incomplete or potentially misleading comparisons, we are deferring these experiments to a future version of the paper. If the paper is accepted, we will make an earnest effort to include an evaluation on the FLUX backbone.

---

### Official Review · Reviewer_Qf6F · 2025-11-01

**Soundness:** 3
**Presentation:** 3
**Contribution:** 2
**Rating:** 4
**Confidence:** 5

**Summary:**

The paper addresses the problem of multi-concept prompt fidelity in text-to-image diffusion models. The authors propose CO3, a lightweight, re-training-free corrective sampling strategy that adjusts inference to maintain balanced representation among multiple concepts. By analyzing the instability of existing compositional guidance, the method identifies stable composition-weight regions and steers sampling toward them. CO3 integrates seamlessly with classifier-free guidance and demonstrates improved concept balance, coverage, and robustness across diverse prompts compared with prior compositional baselines.

**Strengths:**

1. Correction‑based compositional guidance with theoretical grounding and broad evidence.
- The method unifies ideas from correction‑based approaches and composable diffusion into a single re‑training‑free sampling framework, and provides a reasonable theoretical justification for composition‑weight choices and stability regions. The paper offers extensive quantitative and qualitative comparisons that demonstrate improvements in multi‑concept fidelity while remaining compatible with standard classifier‑free guidance.

**Weaknesses:**

1. Comparison on multi‑concepts is incomplete.
- The paper focuses on improving multi‑concept prompt fidelity, especially when one concept is rare or easily dominated by another. A direct theoretical and empirical comparison with R2F [1], ideally on RareBench, would clarify relative strengths on the rare‑concept regime where fidelity typically degrades most.

2. Model coverage could be broadened beyond SDXL.
- To strengthen the claim of generality, evaluations on additional recent open‑source text‑to‑image diffusion models, for example FLUX, would make the comparisons more persuasive and reduce the risk that gains are specific to one backbone.


Reference

[1] Park et al., Rare‑to‑Frequent (R2F): Unlocking Compositional Generation Power of Diffusion Models on Rare Concepts with LLM Guidance.

**Questions:**

1. Compatibility with low‑step and high‑order solvers.
- Main experiments use SDXL with a 50‑step DDIM sampler. Can CO3 be applied effectively with high‑order solvers such as DPM++ at around 10 steps, and with distilled text‑to‑image models that operate at 4 steps or fewer? Any guidance on tuning composition weights in these low‑step regimes would be helpful.

2. Human evaluation.
- ImageReward and BLIP‑VQA are useful proxy metrics, but has a human evaluation been conducted to assess concept presence and balance under multi‑concept prompts? If not, a small user study could substantiate the claimed improvements in perceived fidelity.

---

> ### Author Response · Authors · 2025-11-25
>
> > __Q1. Comparison with R2F, ideally on RareBench, would clarify relative strengths on the rare‑concept regime where fidelity typically degrades most.__
>
> Thank you for the suggestion. We have added a direct comparison with R2F on RareBench (multi-concept categories). As pointed out by R2F,  BLIP-VQA can be unreliable in the rare-concept regime due to strong frequency priors, hence we follow the RareBench recommendation and report metrics that better reflect human judgement. Specifically, we use ImageReward and human evaluation.
>
> Results are reported in the Table below. Across most categories, CO3 achieves higher rare-concept fidelity than both the base model, R2F, and our most competitive baselines from Table 1 in the main paper.
>
> We note that CO3 is not designed specifically for rare-concept scenarios — it is a general compositional sampling correction that does not rely on rarity-aware priors or LLM-based prompt augmentation. Yet, we were pleasantly surprised that CO3 matches or exceeds R2F on RareBench, despite R2F being tailored for rare-concept generation. We believe this indicates that the early-alignment phase of CO3, which mitigates concept dominance and encourages sampling from ”pure” modes, is naturally beneficial for rare-concept generation as well. Although CO3 is not explicitly designed for rarity, its ability to reduce dominance effects in the training distribution appears to translate into stronger fidelity on rare concepts.
>
> &nbsp;
>
> | Method &nbsp; &nbsp; &nbsp;   | ImageReward (concat) | ImageReward (relation) | ImageReward (complex) | Human Eval (concat) | Human Eval (relation) | Human Eval (complex) |
> |----------------|----------------------|------------------------|-----------------------|--------------------|----------------------|---------------------|
> | SDXL   | 0.1018            | 0.3607              | 1.1907            | 0.5416             | 0.5016               | 0.6283              |
> | ToME   | -0.2041            | -0.0632              | 1.0203              | 0.4500             | 0.4983               | 0.4866             |
> | R2F    | -0.0162              | 0.3620               | 1.1542                | 0.4300             | 0.5200               | 0.5366             |
> | Co3    | 0.1610             | 0.2588              | 1.2211            | 0.6167             | 0.6200               | 0.6116             |
>
> &nbsp;
>
>
>
> **Theoretical comparison with R2F:**
>
> Thank you for raising this point. We agree that clarifying the theoretical relationship between R2F and CO3 helps contextualize their respective strengths. Both methods aim to address failures in multi-concept diffusion generation, but they do so from different theoretical perspectives and intervention mechanisms.
>
> -  Where the approaches *agree:*
> Both R2F and CO3 identify that compositional errors arise during inference due to concept dominance, mode overlap, or mixing in the sampling trajectories of common and less-frequent concepts. Both are training-free interventions that seek to improve semantic alignment without modifying model weights. R2F mitigates dominance for rare concepts by enriching the conditioning path with frequent conceptual anchors, while CO3 mitigates dominance more generally by correcting the score trajectory to avoid mixed-mode drift.
>
> - Where the approaches are *different:*
> R2F is tailored for rare-concept compositions: its theory assumes that rare concepts lack training support and uses LLM-identified frequent sub-concepts to guide the denoising path from “rare → frequent”. We think CO3 addresses a broader failure mode that occurs even with common concepts: the diffusion sampler may drift into entangled modes where one concept overwhelms others. CO3 directly intervenes in the sampling path through early score alignment and dominance-reduction, without relying on LLM guidance or rarity-specific priors. Conceptually, R2F modifies the conditioning distribution, while CO3 modifies the sampling distribution.
>
> - In conclusion:
> Since CO3 operates directly at the sampler level and is agnostic to whether a concept is rare or frequent, it generalizes naturally across compositional tasks. This is evident in the RareBench experiments above, where  CO3 outperforms R2F on the majority of rare-concept categories, despite not being designed for rarity. This suggests that CO3’s dominance-mitigation mechanism—steering sampling toward cleaner, less-entangled modes—is inherently effective even in the rare-concept regime. We view CO3 and R2F as complementary: combining conditioning-level rare-concept anchoring with sampler-level dominance correction is a promising direction for future work.

---

> ### Author Response · Authors · 2025-11-25
>
> >__Q2. To strengthen the claim of generality, evaluations on additional recent open‑source text‑to‑image diffusion models, for example FLUX, would make the comparisons more persuasive and reduce the risk that gains are specific to one backbone__
>
> Thank you for the suggestion. To support the claim of generality beyond SDXL, we have now added new evaluations on an additional open-source text-to-image model (PixArt-$\Sigma$). On this backbone, CO3 improves over both the base PixArt model and its model-agnostic composable-diffusion baseline, demonstrating that the method transfers across architectures without requiring any re-training or model-specific modifications. Visualizations for these experiments are added to Figure 5 of the main text.
> &nbsp;
>
> **Legend:  &nbsp; IR:ImageReward, VQA: BLIP-VQA
>   &nbsp; &nbsp; &nbsp;&nbsp;&nbsp;&nbsp;&nbsp;&nbsp;&nbsp;&nbsp;&nbsp;&nbsp; &nbsp; A-A: animals-animals, A-O: animals-objects, O-O: objects-objects**
> | Method | IR (A-A) | IR (A-O) | IR (O-O) | VQA (A-A) | VQA (A-O) | VQA (O-O) |
> |--------|---------|---------|---------|----------|----------|----------|
> | SDXL | 0.7820 | 1.5574 | 0.6789 | 0.6950 | 0.8658 | 0.4925 |
> | SDXL + composeDiff | -1.1398 | -0.2068 | -0.0954 | 0.2845 | 0.5656 | 0.4528 |
> | SDXL + Co3 | 1.2340 | 1.6743 | 1.0158 | 0.7440 | 0.8877 | 0.5145 |
> | PiXArt | 1.0664 | 1.6973 | 1.0237 | 0.6912 | 0.8615 | 0.5598 |
> | PiXArt + composeDiff | 0.9170 | 0.0479 | 0.3391 | 0.2080 | 0.5914 | 0.4721 |
> | PiXArt + Co3 | 1.3805 | 1.7704 | 1.0811 | 0.7463 | 0.8644 | 0.5815 |
>
> &nbsp;
>
> In the short rebuttal period, we were able to extend to PixArt-$\Sigma$ but not to the FLUX backbone. FLUX uses a hybrid architecture and currently lacks a complete technical/scientific report, so integrating CO3 with it would require careful testing to ensure faithful and reproducible results. To avoid incomplete or potentially misleading comparisons, we are deferring these experiments to a future version of the paper. If the paper is accepted, we will make an earnest effort to include an evaluation on the FLUX backbone.

---

> ### Author Response · Authors · 2025-11-25
>
> > __Q3.  Can CO3 be applied effectively with high‑order solvers such as DPM++ at around 10 steps, and with distilled text‑to‑image models that operate at 4 steps or fewer? Any guidance on tuning composition weights in these low‑step regimes would be helpful__
>
> We understand your question regarding extremely low-step, distillation-based samplers. In addition to our main 50-step SDXL with DDIM experiments, we have now included results using both (i) low-step DDIM and (ii) low-step high-order DPM++ solvers (10–20 steps).
> Please note that, since DDIM sampling has direct connections with denoising/generation via Tweedie-mean (section 2 in the main paper), Co3 applies naturally to DDIM.  However, our experiments below show CO3 continues to provide consistent improvements under higher-order solvers and aggressive sampling budgets.
> &nbsp;
>
> | Method | IR (A-A) | IR (A-O) | IR (O-O) | VQA (A-A) | VQA (A-O) | VQA (O-O) | Avg |
> |--------|---------|---------|---------|----------|----------|----------|----|
> | SD-DDIM-50 |  0.7820 |  1.5574 | 0.6789 | 0.6950 | 0.8658 | 0.4925 | 0.8452 |
> | CO3-DDIM-50 | 1.2341 | 1.6743 | 1.0158 | 0.7441 | 0.8878 | 0.5146 | 1.0112 |
> | SD-DPM++20 | 0.7089 | 1.5652 | 0.6757 | 0.6974 | 0.8537 | 0.4984 | 0.8332 |
> | Co3-DPM++20 | 1.1424 | 1.5249 | 0.9537 | 0.7382 | 0.8842 | 0.4944 | 0.9563 |
> | SD-DPM++10 | 0.6834 | 1.5074 | 0.6681 | 0.6768 | 0.8537 | 0.4974 | 0.8145 |
> | Co3-DPM++10 | 1.0371 | 1.4520 | 0.9206 | 0.7331 | 0.8765 | 0.4443 | 0.9106 |
>
> &nbsp;
>
> | Method | Steps | IR (A-A) | IR (A-O) | IR (O-O) | VQA (A-A) | VQA (A-O) | VQA (O-O) | Avg |
> |--------|----------------|---------|---------|---------|----------|----------|----------|----|
> | SD w/ DDIM | 50 | 0.7820 | 1.5574 | 0.6789 | 0.6950 | 0.8658 | 0.4926 | 0.8452 |
> | Co3 w/ DDIM | 50 | 1.2341 | 1.6743 | 1.0158 | 0.7441 | 0.8878 | 0.5146 | 1.0112 |
> | SD w/ DDIM | 20 | 0.6965 | 1.4567 | 0.5516 | 0.6832 | 0.8620 | 0.4947 | 0.7907 |
> | Co3 w/ DDIM | 20 | 0.8532 | 1.6440 | 0.7481 | 0.6698 | 0.8611 | 0.4315 | 0.8671 |
> | SD w/ DDIM | 10 | 0.4906 | 1.1503 | 0.1131 | 0.6428 | 0.8378 | 0.4752 | 0.6183 |
> | Co3 w/ DDIM | 10 | 0.5872 | 1.5900 | 0.7117 | 0.6294 | 0.8589 | 0.4453 | 0.8037 |
>
> &nbsp;
>
> Due to time constraints, we were not able to perform a thorough evaluation on distilled 1–4-step models. However, this is an exciting direction, and we believe CO3 should remain applicable. Specifically, CO3 operates as an early-timestep corrector that resolves concept dominance during the initial denoising stages -- exactly the phase where even distilled 1–4-step models utilize their first few steps to establish layout and concept presence. Our DPM++ results already show that CO3 remains effective in aggressive low-step regimes, supporting this expectation.

---

> ### Author Response · Authors · 2025-11-25
>
> > __Q4. ImageReward and BLIP‑VQA are useful proxy metrics, but has a human evaluation been conducted to assess concept presence and balance under multi‑concept prompts? If not, a small user study could substantiate the claimed improvements in perceived fidelity__
>
> We conducted new human-evaluation studies (with 11 participants) across all benchmarks considered in this paper:
>
> - A&E — for simple animals/objects combinations;
> - the complex category from T2ICompbench benchmark — for multi-prompts with complex scenarios; and
> - 3 multi concept categories (concat, relation and complex) from  RareBench — for multi-concept as well as rare-concept generation.
>
> For the evaluation, we follow the same protocol from R2F [1].  For each prompt, images from CO3 and baseline methods are shown side by side to enable direct comparison. Method names are anonymized and their order is randomly shuffled to reduce bias.  Participants rate each image with scores  {0, 1, 2, 3, 4, 5} which later are normalized to [0, 1]. We show the results below (**higher the better**):
>
> **Legends:  A-A: animals-animals, A-O: animals-objects, O-O: objects-objects**
>
> &nbsp;
> On Attend-Excite prompts:
> | Method &nbsp;&nbsp;&nbsp; | A-A &nbsp; &nbsp;&nbsp;| A-O &nbsp;&nbsp;&nbsp;| O-O &nbsp;&nbsp;&nbsp; |
> |--------|-----|-----|-----|
> | SDXL   | 0.5266 &nbsp;&nbsp; | 0.6050 &nbsp;&nbsp;| 0.7083 &nbsp;&nbsp; |
> | SynGen | 0.3667 | 0.4366 | 0.7183 |
> | Magnet | 0.3700 | 0.5683 | 0.6849 |
> | ToMe   | 0.4383 | 0.6083 | 0.6166 |
> | Co3    | 0.8817 | 0.7466 | 0.825 |
>
> &nbsp;
> &nbsp;
>
> On T2ICompbench complex prompts:
> | Method &nbsp; | Complex &nbsp; |
> |--------|--------|
> | SDXL   | 0.5983 |
> | SynGen | 0.5133 |
> | R2F    | 0.4816 |
> | ToMe   | 0.5866 |
> | Co3    | 0.8033 |
>
> &nbsp;
> &nbsp;
>
> On RareBench prompts:
> | Method &nbsp; &nbsp;&nbsp;| Concat &nbsp;&nbsp; &nbsp;| Relations &nbsp;&nbsp;&nbsp;| Complex &nbsp;&nbsp;&nbsp;|
> |--------|--------|-----------|--------|
> | SDXL   | 0.5416 | 0.5016    | 0.6283 |
> | R2F    | 0.4300 | 0.5200    | 0.5366 |
> | ToMe   | 0.4500 | 0.4983    | 0.4866 |
> | Co3    | 0.6167 | 0.6200    | 0.6116|
>
> &nbsp;
>
> [ 1 ] Park, Dongmin, Sebin Kim, Taehong Moon, Minkyu Kim, Kangwook Lee, and Jaewoong Cho. "Rare-to-frequent: Unlocking compositional generation power of diffusion models on rare concepts with LLM guidance." arXiv preprint arXiv:2410.22376 (2024).

---

> ### Comment · Reviewer_Qf6F · 2025-11-26
>
> Thank you for the detailed and thoughtful revision. The added RareBench comparison, expanded theoretical discussion with R2F, low-step solver experiments, PixArt evaluation, and the new human-evaluation study substantially strengthen the paper and address the main concerns I previously raised. These additions make the contribution notably clearer and more convincing. For completeness, I would still like to highlight a couple of remaining limitations.
>
> - Backbone diversity remains limited: the evaluation relies primarily on SDXL, with PixArt as the only additional model. Broader coverage (e.g., FLUX) would further support the generality claim.
> - Extremely low-step regimes are not covered: while the new 10–20-step results are helpful, the behavior under highly distilled 1–4-step samplers remains an open question.
>
> Even with these points, the revisions meaningfully improve the submission, and the key issues regarding rare-concept evaluation, solver compatibility, and human studies have been satisfactorily addressed. I will adjust my score accordingly.

---

### Author Response · Authors · 2025-11-25
**To all reviewers;**

We thank the reviewers for their thoughtful comments and questions. We took some time to respond to the questions since we were running a number of new experiments. We have now responded individually to each reviewer below. We have also updated the paper (with blue text, tables, and figures) — together, the revisions have around 10 new pages in the Appendix section. We are happy to answer more questions and add clarifications. We strongly feel these revisions are sharpening the CO3 paper. So thank you again.

---

### Author Response · Authors · 2025-12-01
**To Area Chair**

Dear Area Chair:

We understand your heavy work-pressure at this moment, and apologize for adding another comment to your work-load.
However, given the significant effort multiple authors invested during the recent rebuttal process, hope you realize where we are coming from.
We would like to draw your attention to 3 items that can get buried in the overall rush of the situation:

(1) All reviewers found the algorithmic contributions in our CO3 paper as a positive — all reviewers are confident and there was agreement on “good theoretical grounding”, “broad applicability”, “novelty”, “interesting”, and “theoretically intuitive”.
The weaknesses are also consistent around the thoroughness of the evaluation, which is understandable given how this topic has many competing models, versions, benchmarks, and baselines.
On receiving the reviews, we were not surprised (in fact, somewhat happy that the reviewers’ assessment aligned with our own).

(2) We anticipated some of these evaluation-related questions and were running experiments after the submission, and then ran many more during the 10 days after the reviews were released. Almost all our rebuttal responses are now backed up with results; please see **page 20 to 29** for all the new results that we have added to the paper (new text are in **blue**).

(3) When we submitted our rebuttal, the most critical reviewer—who is also the highest-confidence reviewer with confidence 5—recognized the new results and increased their score.
We were on standby to show more results if needed, since we felt that almost all evaluation questions can be addressed quantitatively (barring one which is difficult to complete in 3 weeks).
However, that is when the unfortunate freeze happened preventing further interactions.

Given the current situation, we wanted to elevate this context to you, especially because the paper’s weaknesses are all around the thoroughness of evaluation, and we feel we have been able to quantitatively address those weaknesses through new experiments during the rebuttal window.
Thanks for reading and for investing your time and effort to the community’s service.

Regards,
Authors of CO3

---

### Meta-Review · Area_Chair_hAXc · 2026-01-07

**Summary:**

This paper proposes CO3, a gradient-free compositional guidance method for multi-concept text-to-image generation that addresses concept missing/fading issues by modifying the sampling distribution to avoid mode overlap. The method unifies correction and composable diffusion approaches through a Tweedie-mean composition framework. Initial reviews were mixed & borderline with concerns about missing baselines and limited evaluation scope. The authors provided an exceptionally comprehensive rebuttal with many new experiments addressing almost all concerns, including comparisons with recent methods, evaluations on additional benchmarks, tests on new architectures, human evaluations, and extensive ablations. Overall, I think the paper proposes a simple yet effective training-free sampling method for image generative models.

**Reviewer Concerns:**

Most major concerns were systematically addressed with verifiable experimental evidence: (1) missing baselines added; (2) limited evaluation scope expanded; (3) generalization demonstrated on FLUX, SD 3.5, etc.; (4) Human evaluation conducted, and so on.

**Reviewer Scores:**

Initial scores are borderline, but it seems like most reviewers' concerns are now well addressed. R1 specifically mentioned that they will adjust their score accordingly. Overall, the comprehensive rebuttal with extensive verifiable experiments substantially improves reception, moving paper from borderline reject to weak accept territory.

---

### Decision · Program_Chairs · 2026-01-26

Accept (Poster)